# Finite Element Iterative Methods for the 3D Steady Navier–Stokes Equations [note 1]

**DOI:** 10.3390/e23121659

**Published:** 2021-12-09

**Authors:** Yinnian He

**Affiliations:** School of Mathematics and Statistics, Xi’an Jiaotong University, Xi’an 710049, China; heyn@mail.xjtu.edu.cn

**Keywords:** Navier–Stokes equations, Oseen iterative equations, Newton iterative equations, Stokes iterative equations, weak formulation, finite element, error estimate, discrete inf-sup condition, 76D07, 65N06, 65N30, 65N15

## Abstract

In this work, a finite element (FE) method is discussed for the 3D steady Navier–Stokes equations by using the finite element pair $X_{h}\times M_{h}$. The method consists of transmitting the finite element solution $\left(u_{h},p_{h}\right)$ of the 3D steady Navier–Stokes equations into the finite element solution pairs $\left(u_{h}^{n},p_{h}^{n}\right)$ based on the finite element space pair $X_{h}\times M_{h}$ of the 3D steady linearized Navier–Stokes equations by using the Stokes, Newton and Oseen iterative methods, where the finite element space pair $X_{h}\times M_{h}$ satisfies the discrete inf-sup condition in a 3D domain Ω. Here, we present the weak formulations of the FE method for solving the 3D steady Stokes, Newton and Oseen iterative equations, provide the existence and uniqueness of the FE solution $\left(u_{h}^{n},p_{h}^{n}\right)$ of the 3D steady Stokes, Newton and Oseen iterative equations, and deduce the convergence with respect to (σ,h) of the FE solution $\left(u_{h}^{n},p_{h}^{n}\right)$ to the exact solution (u,p) of the 3D steady Navier–Stokes equations in the $H^{1}-L^{2}$ norm. Finally, we also give the convergence order with respect to (σ,h) of the FE velocity $u_{h}^{n}$ to the exact velocity *u* of the 3D steady Navier–Stokes equations in the $L^{2}$ norm.

## 1. Introduction

The incompressible Navier–Stokes equations reflect the basic mechanical law of viscous fluid flow, which have important implications in fluid mechanics. This problem is one of the main systems studied in pipe flow, flow around airfoils, blood flow, weather and convective heat transfer inside industrial furnaces. Therefore, solving the 3D steady Navier–Stokes equations is of great significance and application value in the field of scientific research and engineering application. Lots of works are devoted to this problem, and the finite element methods, finite volume methods and finite difference methods are the most successful methods. There are many scholars who have studied the numerical methods of the Navier–Stokes equations; see, for example, the monographs of Temam [1], Girault and Raviart [2], Quarteroni and Valli [3], Glowinski [4], Elman et al. [5], Heywood and Rannacher [6,7,8,9], Layton [10], and He et al. [11,12,13,14,15]. An important area that is left out is the development of high order spectral volume and spectral difference methods advanced by Kannan et al. [16,17,18,19,20,21,22,23] and Sun et al. [24]. In recent years, the weak Galerkin method [25] and virtual element method [26,27,28] have also made great contributions to solve the Navier–Stokes equations. Chen et al. in [29] proposed a dimension splitting method for the 3D steady Navier–Stokes equations and in [30], proposed a dimension splitting and characteristic projection method for the 3D time-dependent Navier–Stokes equations, giving some numerical examples to verify the effectiveness of the algorithm. However, the results of the numerical analysis are not given in their papers. Much more numerical methods for the Navier–Stokes equations can be found in [31,32,33,34,35], and the references therein. Despite the considerable increase in the available computing power in recent decades, there are still some difficulties in solving the 3D steady Navier–Stokes equations under the uniqueness condition, that is, how to overcome the divergence free constraint and the nonlinearity of the steady Navier–Stokes equations in the 3D space.

Recently, He and Li [11] and Zhang et al. [36] made a great effort to overcome the difficulties mentioned above in solving the 2D steady Navier–Stokes equations; they used the finite element pair $X_{h}\times M_{h}$, satisfying the discrete inf-sup condition in a 2D domain Ω, which overcomes the difficulty of divergence free constraint, using the Oseen, Newton and Stokes iterative finite element methods to overcome the difficulty of nonlinearity of the steady Navier–Stokes equations in the 2D space.

Furthermore, in order to overcome the difficulties mentioned above in solving the 3D steady Navier–Stokes equations, Xu and He [37] and He [38] used the finite element pair $X_{h}\times M_{h}$, satisfying the discrete inf-sup condition in a 2D/3D domain Ω, which overcomes the difficulty of divergence free constraint, using the Stokes, Newton and Oseen iterative finite element methods to overcome the difficulty of nonlinearity of the steady Navier–Stokes equations in the 2D/3D space. However, in [37,38], they provided some poor stability and convergence results under the strong stability and convergence conditions. For the Stokes iterative finite element method, the stability result is $\nu ∥\tilde{\nabla }u_{h}^{n}{∥}_{0,\Omega }\leq 2{∥F∥}_{-1,\Omega }$ and the convergence result is $\nu ∥\tilde{\nabla }\left(u_{h}^{n}-u_{h}\right){∥}_{0,\Omega }\leq {\left(3\sigma \right)}^{n}{∥F∥}_{-1,\Omega }$ under the strong stability and convergence condition $0<\sigma \leq \frac{1}{4}$. For the Newton iterative finite element method, the stability result is $\nu ∥\tilde{\nabla }u_{h}^{n}{∥}_{0,\Omega }\leq \frac{4}{3}{∥F∥}_{-1,\Omega }$ and the convergence result is $\nu ∥\tilde{\nabla }\left(u_{h}^{n}-u_{h}\right){∥}_{0,\Omega }\leq {\left(\frac{9}{5}\sigma \right)}^{2^{n}-1}{∥F∥}_{-1,\Omega }$ under the strong stability and convergence condition $0<\sigma \leq \frac{1}{3}$.

In this paper, we use the finite element solution $\left(u_{h}^{n},p_{h}^{n}\right)$ of the 3D steady Stokes, Newton and Oseen iterative equations (the 3D steady linearized Navier–Stokes equations) to approximate the solution (u,p) of the 3D steady Navier–Stokes equations. For the Stokes iterative finite element method, the stability result is $\nu ∥\tilde{\nabla }u_{h}^{n}{∥}_{0,\Omega }\leq \frac{6}{5}{∥F∥}_{-1,\Omega }$ and the convergence result is $\nu ∥\tilde{\nabla }\left(u_{h}^{n}-u_{h}\right){∥}_{0,\Omega }\leq {\left(\frac{11}{5}\sigma \right)}^{n}\sigma {∥F∥}_{-1,\Omega }$ under the weak stability and convergence condition $0<\sigma \leq \frac{2}{5}$; for the Newton iterative finite element method, the stability result is $\nu ∥\tilde{\nabla }u_{h}^{n}{∥}_{0,\Omega }\leq \frac{6}{5}{∥F∥}_{-1,\Omega }$ and the convergence result is $\nu ∥\tilde{\nabla }\left(u_{h}^{n}-u_{h}\right){∥}_{0,\Omega }\leq \sigma^{2^{n}}{∥F∥}_{-1,\Omega }$ under the strong stability and convergence condition $0<\sigma \leq \frac{5}{11}$. Compared with the results of [37,38], we obtain better stability and convergence results of the finite element iterative solution $\left(u_{h}^{n},p_{h}^{n}\right)$ of of the 3D steady Navier–Stokes equations under the weak stability and convergence condition.

The paper is structured as follows: some preliminaries on the 3D Navier–Stokes equations are recalled, and the uniform regularity results with respect to ν of the solution (u,p) and the uniqueness condition are reduced in Section 2. The mixed finite element methods for the 3D steady Navier–Stokes equations and the Oseen iterative equations are designed, and the existence, uniqueness and stability of the finite element solution $\left(u_{h},p_{h}\right)$ and $\left(u_{h}^{n},p_{h}^{n}\right)$ on the above equations based on the finite element space pair $X_{h}\times M_{h}$ are proved in Section 3. Moreover, the uniform optimal error estimates of the mixed finite element solution $\left(u_{h},p_{h}\right)$ with respect to the exact solution (u,p) of the 3D steady Navier–Stokes equations is provided in Section 4. The Oseen iterative finite element method is designed, and the uniform optimal error estimates of the Oseen iterative finite element solution $\left(u_{h}^{n},p_{h}^{n}\right)$ with respect to the exact solution (u,p) of the 3D steady Navier–Stokes equations are proven in Section 5. The Newton iterative finite element method is designed, and the uniform optimal error estimates of the Oseen iterative finite element solution $\left(u_{h}^{n},p_{h}^{n}\right)$ with respect to the exact solution (u,p) of the 3D steady Navier–Stokes equations are proven in Section 6. The Stokes iterative finite element method is designed and the uniform optimal error estimates of the Oseen iterative finite element solution $\left(u_{h}^{n},p_{h}^{n}\right)$ with respect to the exact solution (u,p) of the 3D steady Navier–Stokes equations are proven in Section 7. Finally, some conclusions of the Oseen, Newton and Stokes iterative finite element methods are provided in Section 8.

## 2. Preliminaries and the 3D Steady Linearized Navier–Stokes Equations

In this section, we first recall the regularity results on the Stokes equations with the Dirichlet boundary condition in a bounded convex polyhedron $\Omega \subset R^{3}$. Then, we consider the 3D steady Navier–Stokes equations and define the iterative solution $\left(u^{n},p^{n}\right)$ by the 3D steady Oseen iterative equations (the 3D steady linearized Navier–Stokes equations) and obtain the regularity results of the Oseen iterative solution $\left(u^{n},p^{n}\right)$ and the error bound of $\left(u^{n},p^{n}\right)$ to (u,p).

First, we consider the 3D steady Stokes equations in Ω with the Dirichlet boundary condition: (1)$$\begin{array}{ccc} -\tilde{\Delta }u+\tilde{\nabla }p & = & F,\;\;\;\mathrm{in}\Omega , \end{array}$$(2)$$\begin{array}{ccc} \tilde{\nabla }\cdot u & = & 0,\;\;\;\mathrm{in}\Omega , \end{array}$$(3)$$\begin{array}{ccc} u & = & 0,\;\;\;\mathrm{on}\partial \Omega , \end{array}$$
where $u=(u_{1},u_{2},u_{3})$ represents the velocity, *p* the pressure with and $\int_{\Omega }p\left(x,y,z\right)dxdydz=0$, $F=(F_{1},F_{2},F_{3})$ the external volumetric force on the fluid. Additionally, we introduce the following notations: $\tilde{\nabla }=\left(\nabla_{1},\nabla_{2},\nabla_{3}\right)=\left(\partial_{x},\partial_{y},\partial_{z}\right)$, $\tilde{\Delta }=\tilde{\nabla }\cdot \tilde{\nabla }$ and $\tilde{\nabla }u={\left(\nabla_{i}u_{j}\right)}_{3\times 3}$.

Using the Green formula, we deduce the weak formulation of the 3D steady Stokes Equations (Equation 1)–(Equation 3): find (u,p)∈X×M such that for each (v,q)∈X×M there holds
(4)$$\begin{array}{c} G\left(\left(u,p\right),\left(v,q\right)\right)\equiv A\left(u,v\right)-d\left(v,p\right)-d\left(u,q\right)=\tilde{F}\left(v,q\right)\equiv {\left(F,v\right)}_{\Omega }, \end{array}$$
where $X=H_{0}^{1}{\left(\Omega \right)}^{3}$, $M=L_{0}^{2}\left(\Omega \right)$ and $d\left(v,q\right)={\left(\tilde{\nabla }\cdot v,q\right)}_{\Omega }$.

In order to consider the existence and the uniqueness of the solution (u,p)∈X×M, we recall the inf-sup condition of d(v,p) in [1,2].

**Lemma** **1.**
*There exists a positive constant β such that for each p∈M, there exists a $\tilde{u}\in X$ such that*

(5)
$$\begin{array}{c} d\left(\tilde{u},p\right)={∥p∥}_{0,\Omega }^{2},\;\;∥\tilde{\nabla }\tilde{u}{∥}_{0,\Omega }\leq \beta^{-1}{∥p∥}_{0,\Omega }, \end{array}$$

*or*

(6)
$$\begin{array}{c} \underset{v\in X}{\sup}\frac{d(v,p)}{∥\tilde{\nabla }{v∥}_{0,\Omega }}\geq \beta {∥p∥}_{0,\Omega }\;\;\forall p\in M. \end{array}$$


*Next, we need to recall the general Lax–Milgram theorem.*


General Lax–Milgram theorem. For the weak formulation, find (u,p)∈X×M such that for each (v,q)∈X×M, there holds
(7)$$\begin{array}{c} \tilde{G}\left(\left(u,p\right),\left(v,q\right)\right)=\tilde{F}\left(v,q\right), \end{array}$$
if there holds the following condition: (8)$$\begin{array}{ccc} & & |\tilde{F}\left(v,q\right)|\leq ∥\tilde{F}{∥}_{{\left(X\times M\right)}^{\prime }}{(∥v∥}_{X}+{∥q∥}_{M}), \end{array}$$(9)$$\begin{array}{ccc} & & |\tilde{G}{\left(\left(u,p\right),\left(v,q\right)\right)|\leq \alpha (∥u∥}_{X}+{∥p∥}_{M}{)(∥v∥}_{X}+{∥q∥}_{M}), \end{array}$$(10)$$\begin{array}{ccc} & & {\beta (∥u∥}_{X}+{∥p∥}_{M})\leq \underset{(v,q)\in X\times M}{\sup}\frac{\tilde{G}\left(\left(u,p\right),\left(v,q\right)\right)}{{∥v∥}_{X}+{∥q∥}_{M}}, \end{array}$$(11)$$\begin{array}{ccc} & & {\beta (∥v∥}_{X}+{∥q∥}_{M})\leq \underset{(u,p)\in X\times M}{\sup}\frac{\tilde{G}\left(\left(u,p\right),\left(v,q\right)\right)}{{∥u∥}_{X}+{∥p∥}_{M}}, \end{array}$$

Then, (Equation 7) admits a unique solution (u,p)∈X×M, satisfying
(12)$$\begin{array}{ccc} & & {\beta (∥u∥}_{X}+{∥p∥}_{M})\leq ∥\tilde{F}{∥}_{{\left(X\times M\right)}^{\prime }}. \end{array}$$

Now, we give the existence, uniqueness and stability of the solution (u,p)∈X×M for the 3D steady Stokes equations.

**Lemma** **2.**
*If $F\in H^{-1}{\left(\Omega \right)}^{3}$, then (Equation 4) admits a unique solution (u,p)∈X×M, satisfying the following bound:*

(13)
$$\begin{array}{c} ∥\tilde{\nabla }{u∥}_{0,\Omega }\leq {∥F∥}_{-1,\Omega }{,\;\;\beta ∥p∥}_{0,\Omega }\leq 2{∥F∥}_{-1,\Omega }. \end{array}$$



**Proof.** First, we easily prove the following inequalities
(14)$$\begin{array}{c} |\tilde{F}{\left(v,q\right)|=|\left(F,v\right)}_{\Omega }{|\leq ∥F∥}_{-1,\Omega }{∥\tilde{\nabla }v∥}_{0,\Omega }, \end{array}$$
(15)$$\begin{array}{c} G\left(\left(u,p\right),\left(v,q\right)\right)\leq \sqrt{3}(∥\tilde{\nabla }{u∥}_{0,\Omega }+{∥p∥}_{0,\Omega })(∥\tilde{\nabla }{v∥}_{0,\Omega }+{∥q∥}_{0,\Omega }), \end{array}$$
for any (u,p),(v,q)∈X×M. Using Lemma 1, for each (u,p)∈X×M, we set $\varepsilon =\beta^{2}$ and $\left(\tilde{v},\tilde{q}\right)=\left(u-\varepsilon \tilde{u},-p\right)$, where $\tilde{u}$ satisfies
(16)$$\begin{array}{c} d\left(\tilde{u},p\right)={∥p∥}_{0,\Omega }^{2},\;\;∥\tilde{\nabla }\tilde{u}{∥}_{0,\Omega }\leq \beta^{-1}{∥p∥}_{0,\Omega }. \end{array}$$Thus, we have
(17)$$\begin{array}{ccc} \underset{(v,q)\in X\times M}{\sup}\frac{G((u,p),(v,q))}{∥\tilde{\nabla }{v∥}_{0,\Omega }+{∥q∥}_{0,\Omega }} & \geq & \frac{G(\left(u,p\right),\left(\tilde{v},\tilde{q}\right))}{∥\tilde{\nabla }\tilde{v}{∥}_{0,\Omega }+{∥\tilde{q}∥}_{0,\Omega }}\\ & = & \frac{{\left(\tilde{\nabla }u,\tilde{\nabla }\left(u-\varepsilon \tilde{u}\right)\right)}_{\Omega }+\varepsilon d\left(\tilde{u},p\right)}{∥\tilde{\nabla }\left(u-\varepsilon \tilde{u}\right){∥}_{0,\Omega }+{∥p∥}_{0,\Omega }}\\ & \geq & \frac{∥\tilde{\nabla }{u∥}_{0,\Omega }^{2}-\varepsilon \beta^{-1}{∥p∥}_{0,\Omega }∥\tilde{\nabla }{u∥}_{0,\Omega }+\varepsilon {∥p∥}_{0,\Omega }^{2}}{∥\tilde{\nabla }{u∥}_{0,\Omega }+\left(1+\varepsilon \beta^{-1}\right){∥p∥}_{0,\Omega }}\\ & \geq & \frac{0.5∥\tilde{\nabla }{u∥}_{0,\Omega }^{2}+0.5\varepsilon {∥p∥}_{0,\Omega }^{2}}{∥\tilde{\nabla }{u∥}_{0,\Omega }+\left(1+\varepsilon \beta^{-1}\right){∥p∥}_{0,\Omega }}\\ & \geq & \beta_{1}(∥\tilde{\nabla }{u∥}_{0,\Omega }+{∥p∥}_{0,\Omega }). \end{array}$$Since G((u,p),(v,q))=G((v,q),(u,p)), there holds
(18)$$\begin{array}{ccc} & & \underset{(u,p)\in X\times M}{\sup}\frac{G((u,p),(v,q))}{∥\tilde{\nabla }{u∥}_{0,\Omega }+{∥p∥}_{0,\Omega }}=\underset{(u,p)\in X\times M}{\sup}\frac{G((v,q),(u,p))}{∥\tilde{\nabla }{u∥}_{0,\Omega }+{∥p∥}_{0,\Omega }}\\ & \geq & \beta_{1}(∥\tilde{\nabla }{v∥}_{0,\Omega }+{∥q∥}_{0,\Omega }). \end{array}$$Thus, using the general Lax–Milgram theorem, (Equation 4) admits a unique solution (u,p)∈X×M.Taking (v,q)=(u,−p) in (Equation 4), we obtain
(19)$$\begin{array}{c} ∥\tilde{\nabla }{u∥}_{0,\Omega }\leq {∥F∥}_{-1,\Omega }. \end{array}$$Using again Lemma 1 and (Equation 4) with q=0, we have
(20)$$\begin{array}{c} {\beta ∥p∥}_{0,\Omega }\leq ∥\tilde{\nabla }{u∥}_{0,\Omega }+{∥F∥}_{-1,\Omega }\leq 2{∥F∥}_{-1,\Omega }. \end{array}$$Combining (Equation 19) with (Equation 20) yields (Equation 13). The proof ends. □

**Proof.** We introduce the subspace *V* of *X* as follows:
$$\begin{array}{c} V=\{v\in X;d(v,q)=0,\forall q\in M\}. \end{array}$$Thus, we deduce from (Equation 4) that u∈V satisfies
(21)$$\begin{array}{c} A\left(u,v\right)={\left(F,v\right)}_{\Omega },\;\;\forall v\in V, \end{array}$$
where A(u,v) satisfies
(22)$$\begin{array}{c} |A\left(u,v\right)|\leq ∥\tilde{\nabla }{u∥}_{0,\Omega }∥\tilde{\nabla }{v∥}_{0,\Omega },\;\;|A\left(u,u\right)|\geq ∥\tilde{\nabla }{u∥}_{0,\Omega }^{2},\;\;\forall u,\;v\in V. \end{array}$$Using the Lax–Miligram theorem, (Equation 21) admits a unique solution u∈V such that
(23)$$\begin{array}{c} ∥\tilde{\nabla }{u∥}_{0,\Omega }\leq {∥F∥}_{-1,\Omega }, \end{array}$$Now, we introduce a Polar set
$$V^{0}=\left\{g\in X^{\prime };<g,v>=0\;\;\forall v\in V\right\},$$
and define two dual operators Bv∈M and $B^{\prime }q\in X^{\prime }$ such that
$$d\left(v,q\right)={\left(Bv,q\right)}_{\Omega }=<v,B^{\prime }q>\;\;\forall \left(v,q\right)\in X\times M.$$Thus, referring to [1,2], we know that the inf-sup condition (Equation 6) implies that $B^{\prime }$ is a isomorphic operator from *M* onto $V^{0}$. Moreover, we deduce from (Equation 21) that $-\tilde{\Delta }u-F\in V^{0}$. Thus, there exists a unique p∈M such that $-\tilde{\Delta }u-F=B^{\prime }p$ or
(24)$$\begin{array}{c} A\left(u,v\right)-d\left(v,p\right)={\left(F,v\right)}_{\Omega },\;\;\forall v\in X. \end{array}$$Due to u∈V, there holds d(u,q)=0 for each q∈M. Thus, we have proved that (u,p)∈X×M is a unique solution of (Equation 4). Using (Equation 19) and (Equation 20), we show that (u,p)∈X×M satisfies (Equation 13). The proof ends. □

Recalling Temam [1], if $F\in {\left(L^{2}\left(\Omega \right)\right)}^{3}$, there holds the following regularity result of the solution (u,p) for the Stokes equations:(25)$$\begin{array}{c} {∥u∥}_{2,\Omega }+{∥p∥}_{1,\Omega }\leq c_{0}{∥F∥}_{0,\Omega }. \end{array}$$

Next, we consider the 3D steady Navier–Stokes equations with the Dirichlet boundary condition in a bounded domain Ω: (26)$$\begin{array}{ccc} -\nu \tilde{\Delta }u+\tilde{\nabla }p+B\left(u,u\right) & = & F,\;\;\;\mathrm{in}\Omega , \end{array}$$(27)$$\begin{array}{ccc} \tilde{\nabla }\cdot u & = & 0,\;\;\;\mathrm{in}\Omega , \end{array}$$(28)$$\begin{array}{ccc} u & = & 0,\;\;\;\mathrm{on}\partial \Omega , \end{array}$$
with $\int_{\Omega }p\left(x,y,z\right)dxdydz=0$, where $B\left(u,v\right)=\left(u\cdot \tilde{\nabla }\right)v+\frac{1}{2}\tilde{\nabla }\cdot u\;v$.

Using the Green formula, we deduce the weak formulation of the 3D steady Navier–Stokes Equations (Equation 26)–(Equation 28): We find (u,p)∈X×M such that for each (v,q)∈X×M there holds
(29)$$\begin{array}{c} \nu A\left(u,v\right)-d\left(v,p\right)-d\left(u,q\right)+{\left(B\left(u,u\right),v\right)}_{\Omega }={\left(F,v\right)}_{\Omega }, \end{array}$$
where $X=H_{0}^{1}{\left(\Omega \right)}^{3}$, $M=L_{0}^{2}\left(\Omega \right)$.

Here and hereafter, some positive constants *N*, β, $\gamma_{\Omega }$ and $c_{1}$ and some inequalities are stated as follows:(30)$$\begin{array}{ccc} & & {\left(B\left(u,v\right),w\right)}_{\Omega }=0.5{\left(u\cdot \tilde{\nabla }v,w\right)}_{\Omega }-0.5{\left(u\cdot \tilde{\nabla }w,v\right)}_{\Omega }, \end{array}$$(31)$$\begin{array}{ccc} & & {|\left(B\left(u,v\right),w\right)}_{\Omega }|\leq N∥\tilde{\nabla }{u∥}_{0,\Omega }∥\tilde{\nabla }{v∥}_{0,\Omega }{∥\tilde{\nabla }w∥}_{0,\Omega }\;\;\forall u,\;v,\;w\in X,\\ & & {∥B\left(u,v\right)∥}_{0,\Omega }\leq N∥\tilde{\nabla }{u∥}_{0,\Omega }∥\tilde{\nabla }{v∥}_{0,\Omega }^{\frac{1}{2}}{∥v∥}_{2,\Omega }^{\frac{1}{2}},\;\forall u\in X,\;v\in {\left(H_{0}^{1}\left(\Omega \right)\cap H^{2}\left(\Omega \right)\right)}^{3}, \end{array}$$(32)$$\begin{array}{ccc} & & ∥B^{\prime }{\left(v,w\right)∥}_{0,\Omega }\leq N∥\tilde{\nabla }{v∥}_{0,\Omega }∥\tilde{\nabla }{w∥}_{0,\Omega }^{\frac{1}{2}}{∥w∥}_{2,\Omega }^{\frac{1}{2}},\;\forall v\in X,\;w\in {\left(H_{0}^{1}\left(\Omega \right)\cap H^{2}\left(\Omega \right)\right)}^{3}, \end{array}$$(33)$$\begin{array}{ccc} & & {∥\phi ∥}_{L^{2}\left(\Omega \right)}\leq \gamma_{\Omega }{∥\tilde{\nabla }\phi ∥}_{0,\Omega }\;\;\forall \phi \in H_{0}^{1}\left(\Omega \right), \end{array}$$(34)$$\begin{array}{ccc} & & {∥\phi ∥}_{L^{\infty }\left(\Omega \right)}\leq c_{1}∥\tilde{\nabla }{\phi ∥}_{0,\Omega }^{\frac{1}{2}}{∥\phi ∥}_{2,\Omega }^{\frac{1}{2}}\;\;\forall \phi \in \left(H^{2}\left(\Omega \right)\cap H_{0}^{1}\left(\Omega \right)\right), \end{array}$$(35)$$\begin{array}{ccc} & & {∥\phi ∥}_{L^{6}\left(\Omega \right)}\leq c_{1}{∥\tilde{\nabla }\phi ∥}_{0,\Omega }\;\;\forall \phi \in H_{0}^{1}\left(\Omega \right), \end{array}$$
where $N=\left(c_{1}+c_{1}^{\frac{3}{2}}\right)\left(1+\gamma_{\Omega }^{\frac{1}{2}}\right)$.

In fact, using (Equation 33)–(Equation 35) and the Green formula, we have
$$\begin{array}{ccc} & & {\left(B\left(u,v\right),w\right)}_{\Omega }=0.5{\left(u\cdot \tilde{\nabla }v,w\right)}_{\Omega }+0.5{\left(u\cdot \tilde{\nabla }v,w\right)}_{\Omega }+0.5{\left(\tilde{\nabla }\cdot uv,w\right)}_{\Omega }\\ & = & 0.5{\left(u\cdot \tilde{\nabla }v,w\right)}_{\Omega }+0.5\sum_{i,j=1}^{3}{\left({\tilde{\nabla }}_{i}\left(u_{i}v_{j}\right),w_{j}\right)}_{\Omega }\\ & = & 0.5{\left(u\cdot \tilde{\nabla }v,w\right)}_{\Omega }-0.5\sum_{i,j=1}^{3}{\left(u_{i}{\tilde{\nabla }}_{i}w_{j},v_{j}\right)}_{\Omega }\\ & = & 0.5{\left(u\cdot \tilde{\nabla }v,w\right)}_{\Omega }-0.5{\left(u\cdot \tilde{\nabla }w,v\right)}_{\Omega }={\left(u,B^{\prime }\left(v,w\right)\right)}_{\Omega },\\ & & {|\left(B\left(u,v\right),w\right)}_{\Omega }{|\leq 0.5∥u∥}_{L^{3}}(∥\tilde{\nabla }{v∥}_{0,\Omega }{∥w∥}_{L^{6}}+∥\tilde{\nabla }{w∥}_{0,\Omega }{∥v∥}_{L^{6}})\\ & \leq & {0.5∥u∥}_{L^{2}}^{\frac{1}{2}}{∥u∥}_{L^{6}}^{\frac{1}{2}}(∥\tilde{\nabla }{v∥}_{0,\Omega }{∥w∥}_{L^{6}}+∥\tilde{\nabla }{w∥}_{0,\Omega }{∥v∥}_{L^{6}})\\ & \leq & \gamma_{\Omega }^{\frac{1}{2}}c_{1}^{\frac{3}{2}}∥\tilde{\nabla }{u∥}_{0,\Omega }∥\tilde{\nabla }{v∥}_{0,\Omega }{∥\tilde{\nabla }w∥}_{0,\Omega }\;\;\forall u,\;v,\;w\in X,\\ & & {∥B\left(u,v\right)∥}_{0,\Omega }\leq {∥u∥}_{L^{6}}∥\tilde{\nabla }{v∥}_{L^{3}}+∥\tilde{\nabla }{u∥}_{0,\Omega }{∥v∥}_{L^{\infty }}\\ & \leq & {∥u∥}_{L^{6}}∥\tilde{\nabla }{v∥}_{L^{2}}^{\frac{1}{2}}∥\tilde{\nabla }{v∥}_{L^{6}}^{\frac{1}{2}}+c_{1}∥\tilde{\nabla }{u∥}_{0,\Omega }∥\tilde{\nabla }{v∥}_{0,\Omega }^{\frac{1}{2}}{∥v∥}_{2,\Omega }^{\frac{1}{2}}\\ & \leq & \left(c_{1}+c_{1}^{\frac{3}{2}}\right)∥\tilde{\nabla }{u∥}_{0,\Omega }∥\tilde{\nabla }{v∥}_{0,\Omega }^{\frac{1}{2}}{∥v∥}_{2,\Omega }^{\frac{1}{2}},\\ & & ∥B^{\prime }{\left(v,w\right)∥}_{0,\Omega }\leq 0.5(∥\tilde{\nabla }{v∥}_{0,\Omega }{∥w∥}_{L^{\infty }}+∥\tilde{\nabla }{w∥}_{L^{3}}{∥v∥}_{L^{6}})\\ & \leq & 0.5c_{1}∥\tilde{\nabla }{v∥}_{0,\Omega }∥\tilde{\nabla }{w∥}_{L^{2}}^{\frac{1}{2}}{∥w∥}_{2,\Omega }^{\frac{1}{2}}+0.5c_{1}^{\frac{3}{2}}∥\tilde{\nabla }{v∥}_{0,\Omega }∥\tilde{\nabla }{w∥}_{0,\Omega }^{\frac{1}{2}}{∥w∥}_{2,\Omega }^{\frac{1}{2}}\\ & \leq & 0.5\left(c_{1}+c_{1}^{\frac{3}{2}}\right)∥\tilde{\nabla }{u∥}_{0,\Omega }∥\tilde{\nabla }{v∥}_{0,\Omega }^{\frac{1}{2}}{∥v∥}_{2,\Omega }^{\frac{1}{2}}, \end{array}$$
which yield (Equation 30)–(Equation 32).

Now we discuss the existence, uniqueness and regularity results of the solution (u,p) based on (Equation 29).

**Theorem** **1.**
*If $F\in H^{-1}{\left(\Omega \right)}^{3}$ and the uniqueness index $\sigma =\frac{{N∥F∥}_{-1,\Omega }}{\nu^{2}}$ satisfies the uniqueness condition 0<σ<1, then the 3D steady Navier–Stokes equations admit a unique solution (u,p) satisfying the following bound:*

(36)
$$\begin{array}{c} \nu ∥\tilde{\nabla }{u∥}_{0,\Omega }\leq {∥F∥}_{-1,\Omega }{,\;\;\beta ∥p∥}_{0,\Omega }\leq 3{∥F∥}_{-1,\Omega }, \end{array}$$

*and if $F\in L^{2}{\left(\Omega \right)}^{3}$, then there holds the following regularity result:*

(37)
$$\begin{array}{c} {\nu ∥u∥}_{2,\Omega }+{∥p∥}_{1,\Omega }\leq C_{0}{∥F∥}_{0,\Omega }, \end{array}$$

*where $C_{0}=2c_{0}+c_{0}^{2}\gamma_{\Omega }$.*


**Proof.** For the weak formulation (Equation 29), the existence of the solution (u,p) satisfying (Equation 36) can be proved by (Equation 30)–(Equation 31), the uniqueness condition and the Galerkin spectral method referring to [1] or the Galerkin finite element method referring to Section 3. Now, we let $(u^{1},p^{1})\in X\times M$ and $(u^{2},p^{2})\in X\times M$ be the solutions of (Equation 29). Then, $\left(w,\eta \right)=\left(u^{1}-u^{2},p^{1}-p^{2}\right)$ satisfies the following relation
(38)$$\begin{array}{c} \nu A\left(w,v\right)-d\left(v,\eta \right)-d\left(w,q\right)+{\left(B\left(w,u^{1}\right),v\right)}_{\Omega }+{\left(B\left(u^{2},w\right),v\right)}_{\Omega }=0, \end{array}$$By taking (v,q)=(w,−η) in (Equation 38) and using (Equation 30)–(Equation 31) and (Equation 36), we obtain
(39)$$\begin{array}{c} \nu ∥\tilde{\nabla }{w∥}_{0,\Omega }^{2}={\left(B\left(w,u^{1}\right),w\right)}_{\Omega }\leq N∥\tilde{\nabla }u^{1}{∥}_{0,\Omega }∥\tilde{\nabla }{w∥}_{0,\Omega }^{2}\leq \sigma \nu {∥\tilde{\nabla }w∥}_{0,\Omega }^{2}, \end{array}$$
which, with the uniqueness condition 0<σ<1, yields w=0. Using again Lemma 1 and (Equation 38) with w=0, we deduce η=0. Thus, the uniqueness of the solution (u,p) of (Equation 29) is proved. Moreover, if $F\in {\left(L^{2}\left(\Omega \right)\right)}^{3}$, we deduce from (Equation 25), (Equation 32) and (Equation 36) that
(40)$$\begin{array}{ccc} & & {\nu ∥u∥}_{2,\Omega }+{∥p∥}_{1,\Omega }\leq c_{0}{∥F∥}_{0,\Omega }+c_{0}{∥B\left(u,u\right)∥}_{0,\Omega }\\ & \leq & c_{0}{∥F∥}_{0,\Omega }+c_{0}N∥\tilde{\nabla }{u∥}_{0,\Omega }^{\frac{3}{2}}{∥u∥}_{2,\Omega }^{\frac{1}{2}}\\ & \leq & \frac{\nu }{2}{∥u∥}_{2,\Omega }+c_{0}{∥F∥}_{0,\Omega }+0.5c_{0}^{2}N^{2}\nu^{-1}{∥\tilde{\nabla }u∥}_{0,\Omega }^{3}\\ & \leq & \frac{\nu }{2}{∥u∥}_{2,\Omega }+c_{0}{∥F∥}_{0,\Omega }+0.5c_{0}^{2}(N^{2}\nu^{-4}{∥F∥}_{-1,\Omega }^{2}{)∥F∥}_{-1,\Omega }\\ & \leq & \frac{\nu }{2}{∥u∥}_{2,\Omega }+c_{0}{∥F∥}_{0,\Omega }+0.5c_{0}^{2}\gamma_{\Omega }{∥F∥}_{0,\Omega }, \end{array}$$
which yield (Equation 37). The proof ends. □

Setting $u^{-1}=0$, we define the iterative solution $\left(u^{n},p^{n}\right)$ by the 3D steady Oseen iterative equations (the 3D steady linearized Navier–Stokes equations): (41)$$\begin{array}{ccc} -\nu \tilde{\Delta }u^{n}+\tilde{\nabla }p^{n}+B\left(u^{n-1},u^{n}\right) & = & F,\;\;\;\mathrm{in}\Omega , \end{array}$$(42)$$\begin{array}{ccc} \tilde{\nabla }\cdot u^{n} & = & 0,\;\;\;\mathrm{in}\Omega , \end{array}$$(43)$$\begin{array}{ccc} u^{n} & = & 0,\;\;\;\mathrm{on}\partial \Omega , \end{array}$$
where $u^{n}=\left(u_{1}^{n},u_{2}^{n},u_{3}^{n}\right)=\left(w^{n},u_{3}^{n}\right)$. Using the Green formula, we deduce the weak formulation of the 3D steady Oseen iterative Equations (Equation 41)–(Equation 43): we find (u,p)∈X×M such that for each (v,q)∈X×M there holds
(44)$$\begin{array}{c} \nu A\left(u^{n},v\right)-d\left(v,p^{n}\right)-d\left(u^{n},q\right)+{\left(B\left(u^{n-1},u^{n}\right),v\right)}_{\Omega }={\left(F,v\right)}_{\Omega }, \end{array}$$
or
(45)$$\begin{array}{c} G^{n-1}\left(\left(u^{n},p^{n}\right),\left(v,q\right)\right)={\left(F,v\right)}_{\Omega }, \end{array}$$
where
$$\begin{array}{c} G^{n-1}\left(\left(u,p\right),\left(v,q\right)\right)=A^{n-1}\left(u,v\right)-d\left(v,p\right)-d\left(u,q\right),\\ A^{n-1}\left(u,v\right)=\nu A\left(u,v\right)+{\left(B\left(u^{n-1},u\right),v\right)}_{\Omega }. \end{array}$$

In order to prove the existence, uniqueness and stability of the solution $\left(u^{n},p^{n}\right)$ based on (Equation 45), we consider the inf-sup condition of the general bilinear form $G^{n-1}\left(\left(u,p\right),\left(v,q\right)\right)$.

**Lemma** **3.**
*If the bilinear form $A^{n-1}\left(u,v\right)$ satisfies*

(46)
$$\begin{array}{ccc} A^{n-1}\left(u,v\right) & \leq & c_{2}\nu ∥\tilde{\nabla }{u∥}_{0,\Omega }{∥\tilde{\nabla }v∥}_{0,\Omega }, \end{array}$$


(47)
$$\begin{array}{ccc} A^{n-1}\left(u,u\right) & \geq & \nu ∥\tilde{\nabla }{u∥}_{0,\Omega }^{2}, \end{array}$$

*then there exists a $\beta_{1}>0$ such that*

(48)
$$\begin{array}{ccc} & & G^{n-1}\left(\left(u,p\right),\left(v,q\right)\right)\leq c_{2}(\sqrt{\nu }∥\tilde{\nabla }{u∥}_{0,\Omega }+\frac{1}{\sqrt{\nu }}{∥p∥}_{0,\Omega })(\sqrt{\nu }∥\tilde{\nabla }{v∥}_{0,\Omega }+\frac{1}{\sqrt{\nu }}{∥q∥}_{0,\Omega }), \end{array}$$


(49)
$$\begin{array}{ccc} & & \beta_{1}(\sqrt{\nu }∥\tilde{\nabla }{u∥}_{0,\Omega }+\frac{1}{\sqrt{\nu }}{∥p∥}_{0,\Omega })\leq \underset{(v,q)\in X\times M}{\sup}\frac{G^{n-1}\left(\left(u,p\right),\left(v,q\right)\right)}{\sqrt{\nu }∥\tilde{\nabla }{v∥}_{0,\Omega }+\frac{1}{\sqrt{\nu }}{∥q∥}_{0,\Omega }}, \end{array}$$


(50)
$$\begin{array}{ccc} & & \beta_{1}(\sqrt{\nu }∥\tilde{\nabla }{v∥}_{0,\Omega }+\frac{1}{\sqrt{\nu }}{∥q∥}_{0,\Omega })\leq \underset{(u,p)\in X\times M}{\sup}\frac{G^{n-1}\left(\left(u,p\right),\left(v,q\right)\right)}{\sqrt{\nu }∥\tilde{\nabla }{u∥}_{0,\Omega }+\frac{1}{\sqrt{\nu }}{∥p∥}_{0,\Omega }}, \end{array}$$

*where $c_{2}\geq \sqrt{3}$.*


**Proof.** First, using (Equation 46) and (Equation 47), we deduce that $G^{n-1}\left(\left(u,p\right),\left(v,q\right)\right)$ satisfies (Equation 48) and the following inequality
(51)$$\begin{array}{c} G^{n-1}\left(\left(u,p\right),\left(u,-p\right)\right)\geq \nu {∥\tilde{\nabla }u∥}_{0,\Omega }^{2}, \end{array}$$
for any (u,p),(v,q)∈X×M.Using Lemma 1, for each (u,p)∈X×M, we set $\varepsilon =c_{2}^{-2}\beta_{0}^{2}$ and $\left(\tilde{v},\tilde{q}\right)=\left(u-\nu^{-1}\varepsilon \tilde{u},-p\right)$, where $\tilde{u}$ satisfies
(52)$$\begin{array}{c} d\left(\tilde{u},p\right)={∥p∥}_{0,\Omega }^{2},\;\;∥\tilde{\nabla }\tilde{u}{∥}_{0,\Omega }\leq \beta_{0}^{-1}{∥p∥}_{0,\Omega }. \end{array}$$Thus, we have
$$\begin{array}{ccc} \underset{(v,q)\in X\times M}{\sup}\frac{G^{n-1}\left(\left(u,p\right),\left(v,q\right)\right)}{\sqrt{\nu }∥\tilde{\nabla }{v∥}_{0,\Omega }+\frac{1}{\sqrt{\nu }}{∥q∥}_{0,\Omega }} & \geq & \frac{G^{n-1}\left(\left(u,p\right),\left(\tilde{v},\tilde{q}\right)\right)}{\sqrt{\nu }∥\tilde{v}{∥}_{0,\Omega }+\frac{1}{\sqrt{\nu }}{∥\tilde{q}∥}_{0,\Omega }}\\ & \geq & \frac{\nu ∥\tilde{\nabla }{u∥}_{0,\Omega }^{2}-c_{2}\varepsilon ∥\tilde{\nabla }{u∥}_{0,\Omega }{∥\tilde{\nabla }\tilde{u}∥}_{0,\Omega }+\varepsilon \nu^{-1}d\left(\tilde{u},p\right)}{\sqrt{\nu }∥\tilde{\nabla }\left(u-\varepsilon \nu^{-1}\tilde{u}\right){∥}_{0,\Omega }+\frac{1}{\sqrt{\nu }}{∥\tilde{q}∥}_{0,\Omega }}\\ & \geq & \frac{\nu ∥\tilde{\nabla }{u∥}_{0,\Omega }^{2}-c_{2}\varepsilon \beta_{0}^{-1}∥\tilde{\nabla }{u∥}_{0,\Omega }{∥p∥}_{0,\Omega }+\varepsilon \nu^{-1}{∥p∥}_{0,\Omega }^{2}}{\sqrt{\nu }∥\tilde{\nabla }{u∥}_{0,\Omega }+\left(1+\varepsilon \beta_{0}^{-1}\right)\frac{1}{\sqrt{\nu }}{∥p∥}_{0,\Omega }}\\ & \geq & \frac{0.5\nu ∥\tilde{\nabla }{u∥}_{0,\Omega }^{2}+0.5\varepsilon \nu^{-1}{∥p∥}_{0,\Omega }^{2}}{\sqrt{\nu }∥\tilde{\nabla }{u∥}_{0,\Omega }+\left(1+\varepsilon \beta_{0}^{-1}\right)\frac{1}{\sqrt{\nu }}{∥p∥}_{0,\Omega }}\\ & \geq & \beta_{1}(\sqrt{\nu }∥\tilde{\nabla }{u∥}_{0,\Omega }+\frac{1}{\sqrt{\nu }}{∥p∥}_{0,\Omega }), \end{array}$$
which is (Equation 49). Similarly, we can prove (Equation 50). The proof ends. □

Furthermore, we obtain the following regularity and convergence results of the Oseen iterative solution $\left(u^{n},p^{n}\right)$.

**Theorem** **2.**
*If $F\in H^{-1}{\left(\Omega \right)}^{3}$ and 0<σ<1, then (Equation 41)–(Equation 43) admits a unique solution $(u^{n},p^{n})\in X\times M$ such that*

(53)
$$\begin{array}{c} \nu ∥\tilde{\nabla }u^{n}{∥}_{0,\Omega }\leq {∥F∥}_{-1,\Omega },\;\;\beta ∥p^{n}{∥}_{0,\Omega }\leq 3{∥F∥}_{-1,\Omega } \end{array}$$

*and*

(54)
$$\begin{array}{ccc} \nu ∥\tilde{\nabla }\left(u^{n}-u\right){∥}_{0,\Omega }\leq \sigma^{n+1}{∥F∥}_{-1,\Omega },\;\;\beta {∥p^{n}-p∥}_{0,\Omega } & \leq & 3\sigma^{n+1}{∥F∥}_{-1,\Omega }. \end{array}$$


*Furthermore, if $F\in L^{2}{\left(\Omega \right)}^{3}$, then there holds the following regularity result:*

(55)
$$\begin{array}{c} \nu ∥u^{n}{∥}_{2,\Omega }+∥p^{n}{∥}_{1,\Omega }\leq C_{0}{∥F∥}_{0,\Omega }. \end{array}$$



**Proof.** For n=0 and $u^{-1}=0$, we deduce from (Equation 44) that $(u^{0},p^{0})\in X\times M$ satisfies
(56)$$\begin{array}{c} \nu A\left(u^{0},v\right)-d\left(v,p^{0}\right)-d\left(\nu u^{0},q\right)={\left(F,v\right)}_{\Omega }, \end{array}$$
for each (v,q)∈X×M. Using Lemma 2, we show the existence and uniqueness of the solution $\left(u^{0},p^{0}\right)$ satisfying (Equation 53) of (Equation 56). Moreover, using (Equation 25), (Equation 31) and Lemma 1, we easily show that $(u^{0},p^{0})\in X\times M$ satisfies (Equation 54) and (Equation 55). Thus, we show that Theorem 2 holds for n=0.Now, assuming that the conclusions of Theorem 2 hold for n−1, we want to prove that Theorem 2 holds for *n*. Using (Equation 31) and the induction assumption for n−1, we deduce
(57)$$\begin{array}{ccc} & & |A^{n-1}{\left(u,v\right)|\leq \nu ∥u∥}_{X}{∥v∥}_{X}+N∥u^{n-1}{∥}_{X}{∥u∥}_{X}{∥v∥}_{X}\leq {2\nu ∥u∥}_{X}{∥v∥}_{X}, \end{array}$$
(58)$$\begin{array}{ccc} & & A^{n-1}\left(u,u\right)\geq {∥\sqrt{\nu }u∥}_{X}^{2}, \end{array}$$
for any u,v∈X. Using Lemma 2 and the general Lax–Miligram theorem, we deduce that (Equation 41)–(Equation 43) admits a unique solution $(u^{n},p^{n})\in X\times M$ which satisfies
(59)$$\begin{array}{ccc} & & \nu ∥u^{n}{∥}_{X}\leq {∥F∥}_{-1,\Omega }. \end{array}$$Using again Lemma 1, (Equation 59), (Equation 45) with q=0 and the induction assumption for n−1, we deduce
(60)$$\begin{array}{ccc} & & \beta ∥p^{n}{∥}_{0,\Omega }\leq \underset{v\in X}{\sup}\frac{d(v,p^{n})}{{∥v∥}_{X}}\leq {∥F∥}_{-1,\Omega }+\nu ∥u^{n}{∥}_{X}+N∥u^{n-1}{∥}_{X}{∥u^{n}∥}_{X}\\ & \leq & {2∥F∥}_{-1,\Omega }+N\nu^{-2}{∥F∥}_{-1,\Omega }^{2}\leq 3{∥F∥}_{-1,\Omega }. \end{array}$$Thus, (Equation 59) and (Equation 60) yield (Equation 53).Using again (Equation 25), (Equation 32), (Equation 53) and the induction assumptions on n−1, we have
(61)$$\begin{array}{ccc} & & \nu ∥u^{n}{∥}_{2,\Omega }+∥p^{n}{∥}_{1,\Omega }\leq c_{0}{∥F∥}_{0,\Omega }+c_{0}{∥B\left(u^{n-1},u^{n}\right)∥}_{0,\Omega }\\ & \leq & c_{0}{∥F∥}_{0,\Omega }+0.5c_{0}N(∥u^{n-1}{∥}_{X}^{\frac{1}{2}}∥u^{n-1}{∥}_{2,\Omega }^{\frac{1}{2}}∥u^{n}{∥}_{X}+∥u^{n-1}{∥}_{X}∥u^{n}{∥}_{X}^{\frac{1}{2}}∥u^{n}{∥}_{2,\Omega }^{\frac{1}{2}})\\ & \leq & c_{0}{∥F∥}_{0,\Omega }+0.25\nu ∥u^{n}{∥}_{2,\Omega }+0.25\nu {∥u^{n-1}∥}_{2,\Omega }\\ & + & \frac{1}{8}\nu^{-1}N^{2}c_{0}^{2}(∥u^{n-1}{∥}_{X}∥u^{n}{∥}_{X}^{2}+∥u^{n-1}{∥}_{X}^{2}∥u^{n}{∥}_{X})\\ & \leq & c_{0}{∥F∥}_{0,\Omega }+\frac{1}{4}\nu ∥u^{n}{∥}_{2,\Omega }+\frac{1}{4}C_{0}{∥F∥}_{0,\Omega }+\frac{1}{4}\nu^{-4}N^{2}c_{0}^{2}{∥F∥}_{-1,\Omega }^{3}, \end{array}$$
which, with the uniqueness condition, imply (Equation 55).Next, it follows from (Equation 44) and (Equation 29) that
(62)$$\begin{array}{ccc} & & \nu A\left(u^{n}-u,v\right)+{\left(B\left(u^{n-1}-u,u^{n}\right),v\right)}_{\Omega }+{\left(B\left(u,u^{n}-u\right),v\right)}_{\Omega }\\ & & -d\left(v,p^{n}-p\right)-d\left(u^{n}-u,q\right)=0, \end{array}$$Taking $\left(v,q\right)=\left(u^{n}-u,-p^{n}+p\right)\in X\times M$ in (Equation 62) and using (Equation 30) and (Equation 31) yields
$$\begin{array}{ccc} & & \nu ∥u^{n}{-u∥}_{X}\leq N∥u^{n-1}{-u∥}_{X}{∥u^{n}∥}_{X}\\ & \leq & N\nu^{-2}{∥F∥}_{-1,\Omega }\nu ∥u^{n-1}{-u∥}_{X}\leq \sigma \nu {∥u^{n-1}-u∥}_{X}, \end{array}$$
which, with the induction assumption for n−1, yield
(63)$$\begin{array}{ccc} & & \nu ∥u^{n}{-u∥}_{X}\leq \sigma^{n+1}{∥F∥}_{-1,\Omega }. \end{array}$$Finally, using (Equation 31), (Equation 62) and Lemma 1, we obtain
(64)$$\begin{array}{ccc} & & \beta ∥p^{n}{-p∥}_{0,\Omega }\leq \nu ∥u^{n}{-u∥}_{X}+N∥u^{n-1}{-u∥}_{X}∥u^{n}{∥}_{X}+N∥u^{n}{-u∥}_{X}{∥u∥}_{X}\\ & \leq & \nu ∥u^{n}{-u∥}_{X}+N\nu^{-2}{∥F∥}_{-1,\Omega }\nu (∥u^{n}{-u∥}_{X}+\nu ∥u^{n-1}-u{∥}_{X}). \end{array}$$Combining (Equation 63) and (Equation 64) and using the induction assumption for n−1 yields (Equation 54). Hence, Theorem 2 holds for *n*. □

## 3. The Finite Element Method for the 3D Steady Navier-Stokes Equations

In this section, we design a finite element method for the 3D steady Stokes equations, steady Navier–Stokes equations and the Oseen iterative equations. In addition, we provide the existence, uniqueness and stability of the finite element solutions $u_{h}$, $\left(u_{h},p_{h}\right)$ and $\left(u_{h}^{n},p_{h}^{n}\right)$ on the above equations based on the finite element space pair $X_{h}\times M_{h}$.

Let $\tau_{h}=\left\{K\right\}$ be quasi-uniformly regular partition made of tetrahedra with diameters bounded by *h* of Ω. Define the finite element subspaces $S_{h}$ and $S_{h}^{b}$ of $H^{1}\left(\Omega \right)$ based on $P_{1}$ and $P_{1}^{b}$ elements as follows:$$S_{h}=\left\{v_{h}\in C\left(\bar{\Omega }\right)\cap H^{1}\left(\Omega \right);v_{h}{|}_{K}\in P_{1}\left(K\right),\;\;\forall K\in \tau_{h}\right\},$$
$$S_{h}^{b}=\left\{v_{h}\in C\left(\bar{\Omega }\right)\cap H^{1}\left(\Omega \right);v_{h}{|}_{K}\in P_{1}^{b}\left(K\right),\;\;\forall K\in \tau_{h}\right\},$$
where $P_{1}^{b}$ is a bubble element on *K* and satisfies $P_{1}^{b}\left(K\right)=P_{1}\left(K\right)⊕\mathrm{span}\;\hat{b}\left(K\right)$ and $\hat{b}\left(K\right)$ is a bubble function on *K*.

For the 3D steady Stokes equation, Navier–Stokes equations and Oseen iterative equations, we define the finite element subspace pair $X_{h}\times M_{h}$ of X×M as
$$X_{h}={\left(S_{h}^{b}\cap H_{0}^{1}\left(\Omega \right)\right)}^{3},\;\;M_{h}=S_{h}\cap M,$$
$$S_{h}^{b}\cap H_{0}^{1}\left(\Omega \right)=\mathrm{span}\left\{\phi_{1},\phi_{2},\cdots ,\phi_{m}\right\},\;\;M_{h}=S_{h}\cap M=\mathrm{span}\left\{\psi_{1},\psi_{2},\cdots ,\psi_{l}\right\}.$$

**Remark** **1.**
*From [39,40,41], the finite element space pair $X_{h}\times M_{h}$ satisfies the discrete inf-sup condition.*

*We easily deduce that the above finite element spaces satisfy the following standard assumption:*

*There exist the mappings $\pi_{h}\in L\left(X;X_{h}\right)$ such that*

(65)
$$\begin{array}{c} ∥\pi_{h}{v-v∥}_{0,\Omega }+h∥\nabla \left(\pi_{h}v-v\right){∥}_{0,\Omega }\leq c_{3}h^{l}{∥v∥}_{H^{l}},\;\;l=1,2, \end{array}$$

*for each $v\in X\cap {\left(H^{l}\left(\Omega \right)\right)}^{d}$ with d=1 or 3.*

*The $L^{2}$-orthogonal projection operator $\rho_{h}:L^{2}\left(\Omega \right)\to S_{h}$ satisfies:*

(66)
$$∥\phi -\rho_{h}{\phi ∥}_{0,\Omega }\leq c_{3}h^{l}{∥\phi ∥}_{H^{l}}\;\;\forall \phi \in H^{l}\left(\Omega \right),\;\;l=1,\;2.$$


*The inverse inequality holds:*

(67)
$$∥\tilde{\nabla }\phi_{h}{∥}_{0,\Omega }\leq c_{3}h^{-1}{∥\phi_{h}∥}_{0,\Omega }\;\;\forall \phi_{h}\in S_{h}.$$


*There exists a constant $\beta_{0}>0$ such that*

(68)
$$\underset{v_{h}\in X_{h}}{\sup}\frac{d(v_{h},q_{h})}{∥v_{h}{∥}_{X}}\geq \beta_{0}{∥q_{h}∥}_{0,\Omega }\;\;q_{h}\in M_{h}.$$




FE method of the 3D Stokes equations.

Referring to the weak formulation (Equation 4), we design the FE method of the 3D steady Stokes equations as follows: find $\left(u_{h},p_{h}\right)\in X_{h}\times M_{h}$ such that for each $\left(v_{h},q_{h}\right)\in X_{h}\times M_{h}$ there holds
(69)$$\begin{array}{c} A\left(u_{h},v_{h}\right)-d\left(v_{h},p_{h}\right)-d\left(u_{h},q_{h}\right)={\left(F,v_{h}\right)}_{\Omega }. \end{array}$$

**Lemma** **4.**
*If $F\in H^{-1}{\left(\Omega \right)}^{3}$ and the finite element space pair $X_{h}\times M_{h}$ satisfies (Equation 68), then (Equation 69) admits a unique solution $\left(u_{h},p_{h}\right)\in X_{h}\times M_{h}$, satisfying the following bound:*

(70)
$$\begin{array}{c} ∥\tilde{\nabla }u_{h}{∥}_{0,\Omega }\leq {∥F∥}_{-1,\Omega },\;\;\beta_{0}∥p_{h}{∥}_{0,\Omega }\leq 2{∥F∥}_{-1,\Omega }. \end{array}$$



**Proof.** We introduce the subspace $V_{h}$ of $X_{h}$ as follows:
$$\begin{array}{c} V_{h}=\left\{v_{h}\in X_{h};d\left(v_{h},q_{h}\right)=0,\forall q_{h}\in M_{h}\right\}. \end{array}$$Thus, we deduce from (Equation 69) that $u_{h}\in V_{h}$ satisfies
(71)$$\begin{array}{c} A\left(u_{h},v_{h}\right)={\left(F,v_{h}\right)}_{\Omega },\;\;\forall v_{h}\in V_{h}. \end{array}$$Using the Lax–Miligram theorem with $X=V_{h}$, $\tilde{A}\left(u,v\right)=A\left(u_{h},v_{h}\right)$ and (Equation 22), we show that (Equation 69) admits a unique solution $u_{h}\in V_{h}$, satisfying
(72)$$\begin{array}{c} ∥\tilde{\nabla }u_{h}{∥}_{0,\Omega }\leq {∥F∥}_{-1,\Omega }. \end{array}$$Now, we introduce a Polar set
$$V_{h}^{0}=\left\{g\in X_{h}^{\prime };<g,v_{h}>=0\;\;\forall v_{h}\in V_{h}\right\},$$
and define two dual operators $B_{h}v_{h}\in M_{h}$ and $B_{h}^{\prime }q_{h}\in X_{h}^{\prime }$ such that
$$d\left(v_{h},q_{h}\right)={\left(B_{h}v_{h},q_{h}\right)}_{\Omega }=<v_{h},B_{h}^{\prime }q_{h}>\;\;\forall \left(v_{h},q_{h}\right)\in X_{h}\times M_{h}.$$Thus, referring to [1,2], we know that inf-sup condition (Equation 68) implies that $B_{h}^{\prime }$ is a isomorphic operator from $M_{h}$ onto $V_{h}^{0}$. Moreover, we deduce from (Equation 71) that $-{\tilde{\Delta }}_{h}u_{h}-F\in V_{h}^{0}$. Thus, there exists a unique $p_{h}\in M_{h}$ such that $-{\tilde{\Delta }}_{h}u_{h}-F=B_{h}^{\prime }p_{h}$ or
(73)$$\begin{array}{c} A\left(u_{h},v_{h}\right)-d\left(v_{h},p_{h}\right)={\left(F,v_{h}\right)}_{\Omega },\;\;\forall v_{h}\in X_{h}, \end{array}$$
where the discrete Laplace operator $-\Delta_{h}u_{h}\in X_{h}$ is defined as
$$\left(-\Delta_{h}u_{h},v_{h}\right)=A\left(u_{h},v_{h}\right),\;\;\forall v_{h}\in X_{h}.$$Thus, we have proved that $\left(u_{h},p_{h}\right)\in X_{h}\times M_{h}$ is a unique solution of (Equation 69).Using again (Equation 68) and (Equation 69) with q=0, we have
(74)$$\begin{array}{c} \beta_{0}∥p_{h}{∥}_{0,\Omega }\leq ∥\tilde{\nabla }u_{h}{∥}_{0,\Omega }+{∥F∥}_{-1,\Omega }\leq 2{∥F∥}_{-1,\Omega }. \end{array}$$Combining (Equation 74) with (Equation 72) yields (Equation 70).The proof ends. □

Due to (Equation 68), we can consider the weak formulation of the general Stokes equations: find $\left(u_{h},p_{h}\right):X_{h}\times M_{h}$ such that for each $\left(v_{h},q_{h}\right)\in X_{h}\times M_{h}$, there holds
(75)$$\begin{array}{c} \tilde{A}\left(u_{h},v_{h}\right)-d\left(v_{h},p_{h}\right)-d\left(u_{h},q_{h}\right)={\left(F,v_{h}\right)}_{\Omega }. \end{array}$$

**Lemma** **5.**
*If $F\in X^{\prime }$ and $X_{h}\times M_{h}$ satisfies the discrete inf-sup condition (Equation 68), the bilinear form $\tilde{A}\left(u_{h},v_{h}\right)$ satisfies*

(76)
$$\begin{array}{ccc} \tilde{A}\left(u_{h},v_{h}\right) & \leq & c_{4}\nu ∥u_{h}{∥}_{X}{∥v_{h}∥}_{X}, \end{array}$$


(77)
$$\begin{array}{ccc} \tilde{A}\left(v_{h},v_{h}\right) & \geq & c_{5}\nu {∥v_{h}∥}_{X}^{2}, \end{array}$$

*then (Equation 75) admits a unique solution $\left(u_{h},p_{h}\right)\in X_{h}\times M_{h}$ such that*

(78)
$$\begin{array}{ccc} & & c_{5}\nu ∥u_{h}{∥}_{X}\leq {∥F∥}_{-1,\Omega },\;\;\beta_{0}∥p_{h}{∥}_{0,\Omega }\leq \left(c_{4}c_{5}^{-1}+1\right){∥F∥}_{-1,\Omega }, \end{array}$$

*where $c_{4}\geq \sqrt{3}$.*


**Proof.** First, we deduce from (Equation 75) that $u_{h}\in V_{h}$ satisfies
(79)$$\begin{array}{c} \tilde{A}\left(u_{h},v_{h}\right)={\left(F,v_{h}\right)}_{\Omega },\;\;\forall v_{h}\in V_{h}, \end{array}$$
or
(80)$$\begin{array}{c} \left({\tilde{A}}_{h}u_{h},v_{h}\right)={\left(F,v_{h}\right)}_{\Omega },\;\;\forall v_{h}\in V_{h}. \end{array}$$Using (Equation 76) and (Equation 77) and the Lax–Miligram theorem with $X=V_{h}$ and $\tilde{A}\left(u,v\right)=\tilde{A}\left(u_{h},v_{h}\right)$, we show that (Equation 79) or (Equation 80) admits a unique solution $u_{h}\in V_{h}$ satisfying
(81)$$\begin{array}{c} c_{5}\nu ∥u_{h}{∥}_{X}\leq {∥F∥}_{-1,\Omega }. \end{array}$$Next, (Equation 80) shows ${\tilde{A}}_{h}u_{h}-F\in V_{h}^{0}$. Thus, referring to [1,2], the inf-sup condition (Equation 68) implies that $B_{h}^{\prime }$ is an isomorphic operator from $M_{h}$ onto $V_{h}^{0}$. Thus, there exists a unique $p_{h}\in M_{h}$ such that $-{\tilde{A}}_{h}u_{h}-F=B_{h}^{\prime }p_{h}$ or
(82)$$\begin{array}{c} \tilde{A}\left(u_{h},v_{h}\right)-d\left(v_{h},p_{h}\right)={\left(F,v_{h}\right)}_{\Omega },\;\;\forall v_{h}\in X_{h}. \end{array}$$Thus, we have proved that $\left(u_{h},p_{h}\right)\in X_{h}\times M_{h}$ is a unique solution of (Equation 75).Using again (Equation 68), (Equation 76) and (Equation 75) with $q_{h}=0$, we have
(83)$$\begin{array}{c} \beta_{0}∥p_{h}{∥}_{0,\Omega }\leq c_{4}\nu ∥u_{h}{∥}_{X}+{∥F∥}_{-1,\Omega }\leq \left(1+c_{4}c_{5}^{-1}\right){∥F∥}_{-1,\Omega }. \end{array}$$Combining (Equation 83) with (Equation 81) yields (Equation 78).The proof ends. □

FE method of the 3D Navier-Stokes equations.

Referring to the weak formulation (Equation 29), we design the FE method of the 3D steady Navier–Stokes equations as follows: find $\left(u_{h},p_{h}\right)\in X_{h}\times M_{h}$ such that for each $\left(v_{h},q_{h}\right)\in X_{h}\times M_{h}$, there holds
(84)$$\begin{array}{c} \nu A\left(u_{h},v_{h}\right)-d\left(v_{h},p_{h}\right)-d\left(u_{h},q_{h}\right)+{\left(B\left(u_{h},u_{h}\right),v_{h}\right)}_{\Omega }={\left(F,v_{h}\right)}_{\Omega }. \end{array}$$

**Lemma** **6.**
*If $F\in X^{\prime }$, the finite element space pair $X_{h}\times M_{h}$ satisfies (Equation 68) and 0<σ<1, then (Equation 84) admits a unique solution $\left(u_{h},p_{h}\right)\in X_{h}\times M_{h}$ satisfying the following bound:*

(85)
$$\begin{array}{c} \nu ∥\tilde{\nabla }u_{h}{∥}_{0,\Omega }\leq {∥F∥}_{-1,\Omega },\;\;\beta_{0}∥p_{h}{∥}_{0,\Omega }\leq 3{∥F∥}_{-1,\Omega }. \end{array}$$



**Proof.** We set a bounded convex subset *K* of $X_{h}\times M_{h}$ as
(86)$$\begin{array}{c} K=\{\left(v_{h},q_{h}\right)\in X_{h}\times M_{h};\nu ∥v_{h}{∥}_{0,\Omega }\leq ∥F{∥}_{-1,\Omega },\;\;\beta_{0}∥q_{h}{∥}_{M}\leq 3∥F{∥}_{-1,\Omega }\}, \end{array}$$
and define a map $T:K\to X_{h}\times M_{h}$ such that for each $(w_{h},r_{h})\in K$, $\left(u_{h},p_{h}\right)=T\left(w_{h},r_{h}\right)\in X_{h}\times M_{h}$ satisfies
(87)$$\begin{array}{c} \nu A\left(u_{h},v_{h}\right)+{\left(B\left(w_{h},u_{h}\right),v_{h}\right)}_{\Omega }-d\left(v_{h},p_{h}\right)-d\left(u_{h},q_{h}\right)={\left(F,v_{h}\right)}_{\Omega },\;\;\forall \left(v_{h},q_{h}\right)\in X_{h}\times M_{h}. \end{array}$$
or
(88)$$\begin{array}{c} \tilde{A}\left(w_{h};u_{h},v_{h}\right)-d\left(v_{h},p_{h}\right)-d\left(u_{h},q_{h}\right)={\left(F,v_{h}\right)}_{\Omega },\;\;\forall v_{h}\in V_{h}. \end{array}$$
where
$$\begin{array}{c} \tilde{A}\left(w_{h};u_{h},v_{h}\right)=\nu A\left(u_{h},v_{h}\right)+{\left(B\left(w_{h},u_{h}\right),v_{h}\right)}_{\Omega }. \end{array}$$Due to $(w_{h},r_{h})\in K$, we deduce from (Equation 30) and (Equation 31) and the uniqueness condition that $\tilde{A}\left(w_{h};u_{h},v_{h}\right)$ satisfies
(89)$$\begin{array}{ccc} & & |\tilde{A}\left(w_{h};u_{h},v_{h}\right)|\leq \nu |A\left(u_{h},v_{h}\right)|+N∥w_{h}{∥}_{X}∥u_{h}{∥}_{X}∥v_{h}{∥}_{X}\leq 2\nu ∥u_{h}{∥}_{X}{∥v_{h}∥}_{X}, \end{array}$$
(90)$$\begin{array}{ccc} & & \tilde{A}\left(w_{h};u_{h},u_{h}\right)=\nu {∥u_{h}∥}_{X}^{2}, \end{array}$$
for each $u_{h},\;v_{h}\in X_{h}$. Using Lemma 5, we show that (Equation 87) or (Equation 88) admits a unique solution $\left(u_{h},p_{h}\right)$ satisfying
(91)$$\begin{array}{c} \nu ∥u_{h}{∥}_{X}\leq {∥F∥}_{-1,\Omega },\;\;\beta_{0}∥p_{h}{∥}_{0,\Omega }\leq 3{∥F∥}_{-1,\Omega }. \end{array}$$Thus, (Equation 91) shows that *T* is a map from *K* into *K*. Using the fixed point theorem in finite dimensional space, the map *T* at least has a fixed point $(u_{h},p_{h})\in K$ such that $T\left(u_{h},p_{h}\right)=\left(u_{h},p_{h}\right)$ or (Equation 84) at least admits a solution $(u_{h},p_{h})\in K$. Now, we assume that $(u_{h}^{1},p_{h}^{1})\in K$ and $(u_{h}^{2},p_{h}^{2})\in K$ satisfy (Equation 84). Then $\left(w_{h},r_{h}\right)=\left(u_{h}^{1}-u_{h}^{2},p_{h}^{1}-p_{h}^{2}\right)$ satisfies
(92)$$\begin{array}{c} \nu A\left(w_{h},v_{h}\right)+{\left(B\left(w_{h},u_{h}^{1}\right),v_{h}\right)}_{\Omega }+{\left(B\left(u_{h}^{2},w_{h}\right),v_{h}\right)}_{\Omega }-d\left(v_{h},r_{h}\right)-d\left(w_{h},q_{h}\right)=0,\;\;\forall \left(v_{h},q_{h}\right)\in X_{h}\times M_{h}. \end{array}$$
for each $\left(v_{h},q_{h}\right)\in X_{h}\times M_{h}$. Taking $\left(v_{h},q_{h}\right)=\left(w_{h},-r_{h}\right)$ in (Equation 92) and using (Equation 30) and (Equation 31), we deduce
(93)$$\begin{array}{c} \nu ∥w_{h}{∥}_{X}^{2}\leq N∥w_{h}{∥}_{0,\Omega }^{2}∥u_{h}^{1}{∥}_{X}\leq \sigma \nu {∥w_{h}∥}_{X}^{2}. \end{array}$$Thanks to 0<σ<1, (Equation 93) yields $w_{h}=0$ or $u_{h}^{1}=u_{h}^{2}$. Next, using (Equation 68) and (Equation 93) with $q_{h}=0$, we deduce $r_{h}=0$. The proof ends. □

**Lemma** **7.**
*If $F\in H^{-1}{\left(\Omega \right)}^{3}$, 0<σ<1, the finite element space pair $X_{h_{i}}\times M_{h_{i}}$ is dense in X×M, satisfies (Equation 68) and $X_{h_{i}}\times M_{h_{i}}\subset X_{h_{i+1}}\times M_{h_{i+1}}$ with $\lim_{i\to \infty }h_{i}=0$, then the finite element solutions $\left(u_{h_{i}},p_{h_{i}}\right)\in X_{h_{i}}\times M_{h_{i}}$ based on (Equation 84) satisfying*

(94)
$$\begin{array}{c} \left(u_{h_{i}},p_{h_{i}}\right)is\;weak\;c\mathrm{on}vergent\;to\left(u,p\right)\mathrm{in}\;X\times M\;as\;i\to \infty , \end{array}$$

*here (u,p)∈X×M satisfies (Equation 29).*


**Proof.** We deduce from Lemma 5 that for each 0<h<1 (Equation 84) admits a unique solution $\left(u_{h},p_{h}\right)\in X_{h}\times M_{h}$ satisfying (Equation 85). Applying the compact theorem in X×M, there exist a sequence $\left\{h_{i}\right\}$ and (u,p) such that (Equation 94) holds. Thanks to $X\subset {\left(L^{2}\left(\Omega \right)\right)}^{3}$ being compact, $u_{h_{i}}$ is strong convergent to *u* in ${\left(L^{2}\left(\Omega \right)\right)}^{3}$.Thus, for a fixed $i_{0}$ and $i\geq i_{0}$, there hold
(95)$$\begin{array}{ccc} & & {\left(B\left(u_{h_{i}},u_{h_{i}}\right),v_{h_{i_{0}}}\right)}_{\Omega }-{\left(B\left(u,u\right),v_{h_{i_{0}}}\right)}_{\Omega }\\ & & ={\left(B\left(u_{h_{i}}-u,u_{h_{i}}\right),v_{h_{i_{0}}}\right)}_{\Omega }+{\left(B\left(u,u_{h_{i}}-u\right),v_{h_{i_{0}}}\right)}_{\Omega }\\ & & \leq 0.5∥u_{h_{i}}{-u∥}_{L^{3}\left(\Omega \right)}(∥\tilde{\nabla }u_{h_{i}}{∥}_{0,\Omega }∥v_{h_{i_{0}}}{∥}_{L^{6}\left(\Omega \right)}+∥u_{h_{i}}{∥}_{L^{6}\left(\Omega \right)}∥\tilde{\nabla }v_{h_{i_{0}}}{∥}_{0,\Omega })\\ & & +{(∥u∥}_{L^{6}\left(\Omega \right)}∥\tilde{\nabla }v_{h_{i_{0}}}{∥}_{0,\Omega }+∥\tilde{\nabla }{u∥}_{0,\Omega }∥v_{h_{i_{0}}}{∥}_{L^{6}\left(\Omega \right)})∥u_{h_{i}}{-u∥}_{L^{3}\left(\Omega \right)}\\ & & \leq c_{1}^{\frac{3}{2}}∥u_{h_{i}}{-u∥}_{L^{2}\left(\Omega \right)}^{\frac{1}{2}}∥\tilde{\nabla }\left(u_{h_{i}}-u\right){∥}_{0,\Omega }^{\frac{1}{2}}∥\tilde{\nabla }u_{h_{i}}{∥}_{0,\Omega }{∥\tilde{\nabla }v_{h_{i_{0}}}∥}_{0,\Omega }\\ & & +2c_{1}^{\frac{3}{2}}∥u_{h_{i}}{-u∥}_{L^{2}\left(\Omega \right)}^{\frac{1}{2}}∥\tilde{\nabla }\left(u_{h_{i}}-u\right){∥}_{0,\Omega }^{\frac{1}{2}}∥\tilde{\nabla }{u∥}_{0,\Omega }{∥\tilde{\nabla }v_{h_{i_{0}}}∥}_{0,\Omega }. \end{array}$$Thus, using (Equation 94) and (Equation 95), we deduce
(96)$$\begin{array}{ccc} & & \lim_{i\to \infty }\nu A\left(u_{h_{i}},v_{h_{i_{0}}}\right)=\nu A\left(u,v_{h_{i_{0}}}\right), \end{array}$$
(97)$$\begin{array}{ccc} & & \lim_{i\to \infty }d\left(v_{h_{i_{0}}},p_{h_{i}}\right)=d\left(v_{h_{i_{0}}},p\right), \end{array}$$
(98)$$\begin{array}{ccc} & & \lim_{i\to \infty }d\left(u_{h_{i}},q_{h_{i_{0}}}\right)=d\left(u,q_{h_{i_{0}}}\right), \end{array}$$
(99)$$\begin{array}{ccc} & & \lim_{i\to \infty }{\left(B\left(u_{h_{i}},u_{h_{i}}\right),v_{h_{i_{0}}}\right)}_{\Omega }={\left(B\left(u,u\right),v_{h_{i_{0}}}\right)}_{\Omega }. \end{array}$$Setting $\left(u_{h},p_{h}\right)=\left(u_{h_{i}},p_{h_{i}}\right)\in X_{h_{i}}\times M_{h_{i}}$ and $\left(v_{h},q_{h}\right)=\left(v_{h_{i_{0}}},q_{h_{i_{0}}}\right)\in X_{h_{i_{0}}}\times M_{h_{i_{0}}}$ in (Equation 84), we obtain
(100)$$\begin{array}{c} \nu A\left(u_{h_{i}},v_{h_{i_{0}}}\right)-d\left(v_{h_{i_{0}}},p_{h_{i}}\right)-d\left(u_{h_{i}},q_{h_{i_{0}}}\right)+{\left(B\left(u_{h_{i}},u_{h_{i}}\right),v_{h_{i_{0}}}\right)}_{\Omega }={\left(F,v_{h_{i_{0}}}\right)}_{\Omega }. \end{array}$$Setting i→∞ in (Equation 100) and using (Equation 96)–(Equation 99), we obtain
(101)$$\begin{array}{c} \nu A\left(u,v_{h_{i_{0}}}\right)-d\left(v_{h_{i_{0}}},p\right)-d\left(u,q_{h_{i_{0}}}\right)+{\left(B\left(u,u\right),v_{h_{i_{0}}}\right)}_{\Omega }={\left(F,v_{h_{i_{0}}}\right)}_{\Omega }. \end{array}$$Since $X_{h_{i_{0}}}\times M_{h_{i_{0}}}$ is dense in X×M, setting $i_{0}\to \infty$ in (Equation 101), we deduce that (u,p)∈X×M satisfies (Equation 29).The proof ends. □

## 4. Uniform Error Estimates of FE Solutions

In this section, we provide the error estimate of $\left(u_{h},p_{h}\right)$ with respect to (u,p).

For the FE solution $\left(u_{h},p_{h}\right)\in X_{h}\times M_{h}$ of the Stokes equations, there hold the following error estimates.

**Lemma** **8.**
*If $F\in L^{2}\left(\Omega \right)$ and $X_{h}$ satisfy (Equation 65), then the FE solution $\left(u_{h},p_{h}\right)$ of the Stokes equations satisfies the following error estimates:*

(102)
$$\begin{array}{c} ∥u-u_{h}{∥}_{0,\Omega }+h∥u-u_{h}{∥}_{X}\leq Ch^{2}{∥F∥}_{0,\Omega },\;\;\beta_{0}∥p-p_{h}{∥}_{0,\Omega }\leq Ch{∥F∥}_{0,\Omega }. \end{array}$$



**Proof.** First, we deduce from (Equation 4) and (Equation 69) that
(103)$$\begin{array}{ccc} & & A\left(u-u_{h},v_{h}\right)-d\left(v_{h},p-p_{h}\right)-d\left(u-u_{h},q_{h}\right)=0\;\;\forall \left(v_{h},q_{h}\right)\in X_{h}\times M_{h}. \end{array}$$Using (Equation 103) with $q_{h}=0$ and (Equation 68), we deduce
(104)$$\begin{array}{ccc} & & \beta_{0}∥\rho_{h}p-p_{h}{∥}_{0,\Omega }\leq ∥u-u_{h}{∥}_{X}+\sqrt{3}{∥p-\rho_{h}p∥}_{0,\Omega }. \end{array}$$Next, taking $\left(v_{h},q_{h}\right)=\left(\pi_{h}u-u_{h},-\rho_{h}p+p_{h}\right)$ in (Equation 103), using (Equation 104), (Equation 65) and (Equation 66) and (Equation 25), we deduce
(105)$$\begin{array}{ccc} & & \frac{1}{2}∥u-u_{h}{∥}_{X}^{2}+\frac{1}{2}∥\pi_{h}u-u_{h}{∥}_{X}^{2}=\frac{1}{2}{∥\pi_{h}u-u∥}_{X}^{2}+d\left(\pi_{h}u-u_{h},p-\rho_{h}p\right)+d\left(u-\pi_{h}u,\rho_{h}p-p_{h}\right)\\ & \leq & \frac{1}{2}∥\pi_{h}{u-u∥}_{X}^{2}+\sqrt{3}(∥\pi_{h}u-u_{h}{∥}_{X}∥p-\rho_{h}{p∥}_{0,\Omega }+∥u-\pi_{h}{u∥}_{X}∥\rho_{h}p-p_{h}{∥}_{0,\Omega })\\ & \leq & \frac{1}{2}∥\pi_{h}{u-u∥}_{X}^{2}+\sqrt{3}∥\pi_{h}u-u_{h}{∥}_{X}{∥p-\rho_{h}p∥}_{0,\Omega }\\ & + & \sqrt{3}\beta_{0}^{-1}∥u-\pi_{h}{u∥}_{X}(∥u-u_{h}{∥}_{X}+\sqrt{3}∥\rho_{h}p-p{∥}_{0,\Omega })\\ & \leq & \frac{1}{2}∥\pi_{h}u-u_{h}{∥}_{X}^{2}+\frac{1}{4}∥u-u_{h}{∥}_{X}^{2}+c(∥p-\rho_{h}{p∥}_{0,\Omega }^{2}+∥u-\pi_{h}u{∥}_{X}^{2})\\ & \leq & \frac{1}{2}∥\pi_{h}u-u_{h}{∥}_{X}^{2}+\frac{1}{4}∥u-u_{h}{∥}_{X}^{2}+ch^{2}{(∥u∥}_{2,\Omega }^{2}+{∥p∥}_{1,\Omega }^{2})\\ & \leq & \frac{1}{2}∥\pi_{h}u-u_{h}{∥}_{X}^{2}+\frac{1}{4}∥u-u_{h}{∥}_{X}^{2}+ch^{2}{∥F∥}_{0,\Omega }^{2}. \end{array}$$Combining (Equation 104) and (Equation 105) and (Equation 66), we deduce
(106)$$\begin{array}{c} ∥u-u_{h}{∥}_{X}+\beta_{0}∥p-p_{h}{∥}_{0,\Omega }\leq Ch{∥F∥}_{0,\Omega }. \end{array}$$In order to estimate the $L^{2}\left(\Omega \right)$ bound of the error estimate $u-u_{h}$, we consider the dual equation of the 3D Stokes equations with the Dirichlet boundary condition
(107)$$\begin{array}{ccc} -\tilde{\Delta }\phi +\tilde{\nabla }\psi & = & u-u_{h},\;\;\;\mathrm{in}\Omega , \end{array}$$
(108)$$\begin{array}{ccc} \tilde{\nabla }\cdot \phi & = & 0,\;\;\;\;\mathrm{in}\Omega , \end{array}$$
(109)$$\begin{array}{ccc} \phi & = & 0,\;\;\;\;\mathrm{on}\partial \Omega . \end{array}$$Using the Green formula, we deduce the weak formulation of dual Equations (Equation 107) and (Equation 109): find ϕ∈X such that for each v∈X there holds
(110)$$\begin{array}{c} A\left(\phi ,v\right)-d\left(v,\psi \right)-d\left(\phi ,q\right)={\left(u-u_{h},v\right)}_{\Omega }. \end{array}$$Using again (Equation 25), we have
(111)$$\begin{array}{c} {∥\phi ∥}_{2,\Omega }+{∥\psi ∥}_{1,\Omega }\leq c_{0}{∥u-u_{h}∥}_{0,\Omega }. \end{array}$$Taking $\left(v,q\right)=\left(u-u_{h},p-p_{h}\right)$ in (Equation 110), using (Equation 103) and (Equation 65) and (Equation 66) and (Equation 111), we deduce
(112)$$\begin{array}{ccc} & & ∥u-u_{h}{∥}_{0,\Omega }^{2}=A\left(u-u_{h},\phi \right)-d\left(u-u_{h},\psi \right)-d\left(\phi ,p-p_{h}\right)\\ & = & A\left(u-u_{h},\phi -\pi_{h}\phi \right)-d\left(u-u_{h},\psi -\rho_{h}\psi \right)-d\left(\phi -\pi_{h}\phi ,p-p_{h}\right)\\ & \leq & ∥u-u_{h}{∥}_{X}∥\phi -\pi_{h}{\phi ∥}_{X}+\sqrt{3}(∥u-u_{h}{∥}_{X}∥\psi -\rho_{h}{\psi ∥}_{0,\Omega }+∥\phi -\pi_{h}{\phi ∥}_{X}∥p-p_{h}{∥}_{0,\Omega })\\ & \leq & ch∥u-u_{h}{∥}_{X}{∥\phi ∥}_{2,\Omega }+ch∥u-u_{h}{∥}_{X}{∥\psi ∥}_{1,\Omega }+{ch∥\phi ∥}_{2,\Omega }{∥p-p_{h}∥}_{0,\Omega }\\ & \leq & ch(∥u-u_{h}{∥}_{X}+∥p-p_{h}{∥}_{0,\Omega }{)(∥\phi ∥}_{2,\Omega }+{∥\psi ∥}_{1,\Omega })\\ & \leq & ch(∥u-u_{h}{∥}_{X}+∥p-p_{h}{∥}_{0,\Omega })∥u-u_{h}{∥}_{0,\Omega }. \end{array}$$Combining (Equation 112) with (Equation 106), we obtain (Equation 102).The proof ends. □

Thanks to (Equation 68), we can define the Stokes projection $\left(R_{h},Q_{h}\right):V\times M\to X_{h}\times M_{h}$ such that for each (u,p)∈V×M, $\left(R_{h},Q_{h}\right)=\left(R_{h}\left(u,p\right),Q_{h}\left(u,p\right)\right)\in X_{h}\times M_{h}$ satisfies
(113)$$\begin{array}{c} {\left(\nabla R_{h},\nabla v_{h}\right)}_{\Omega }-d\left(v_{h},Q_{h}\right)-d\left(R_{h},q_{h}\right)={\left(\nabla u,\nabla v_{h}\right)}_{\Omega }-d\left(v_{h},p\right)-d\left(u,q_{h}\right), \end{array}$$
for each $\left(v_{h},q_{h}\right)\in X_{h}\times M_{h}$. From Lemma 4 and Lemma 8, we have
(114)$$\begin{array}{ccc} & & ∥R_{h}{∥}_{X}\leq \sqrt{3}{(∥u∥}_{X}+{∥p∥}_{0,\Omega }),\beta_{0}∥Q_{h}{∥}_{0,\Omega }\leq \left(1+\sqrt{3}\right){(∥u∥}_{X}+{∥p∥}_{0,\Omega })\;\;\forall \left(u,p\right)\in V\times M,\\ & & ∥R_{h}{\left(u,p\right)-u∥}_{0,\Omega }+h∥R_{h}{\left(u,p\right)-u∥}_{X}+h\beta_{0}{∥Q_{h}\left(u,p\right)-p∥}_{0,\Omega } \end{array}$$
(115)$$\begin{array}{ccc} & \leq & c_{6}h^{2}{(∥u∥}_{2,\Omega }+{∥p∥}_{1,\Omega })\;\;\forall \left(u,p\right)\in \left(V\cap {\left(H^{2}\left(\Omega \right)\right)}^{3}\right)\times \left(M\cap H^{1}\left(\Omega \right)\right). \end{array}$$

For the FE solution $\left(u_{h},p_{h}\right)\in X_{h}\times M_{h}$ of the Navier–Stokes equations, there hold the following error estimates.

**Lemma** **9.**
*If $F\in L^{2}\left(\Omega \right)$ and $X_{h}\times M_{h}$ satisfies (Equation 65)–(Equation 68), 0<σ<1 and $\sigma h^{\frac{1}{2}}\leq 1-\sigma$, then the FE solution $\left(u_{h},p_{h}\right)$ satisfies the following error estimates:*

(116)
$$\begin{array}{c} \nu ∥u-u_{h}{∥}_{X}\leq 2c_{6}C_{0}{h∥F∥}_{0,\Omega },\;\beta_{0}∥p-p_{h}{∥}_{0,\Omega }\leq 4c_{6}C_{0}{h∥F∥}_{0,\Omega },\;\nu ∥u-u_{h}{∥}_{0,\Omega }\leq \frac{C}{1-\sigma }h^{2}{∥F∥}_{0,\Omega }. \end{array}$$



**Proof.** First, we deduce from (Equation 4), (Equation 69) and (Equation 113) that
(117)$$\begin{array}{ccc} & & \nu A\left(R_{h}-u_{h},v_{h}\right)-d\left(v_{h},Q_{h}-p_{h}\right)-d\left(R_{h}-u_{h},q_{h}\right)+{\left(B\left(u-u_{h},u\right),v_{h}\right)}_{\Omega }+{\left(B\left(u_{h},u-u_{h}\right),v_{h}\right)}_{\Omega }=0, \end{array}$$
or
(118)$$\begin{array}{ccc} & & \nu A\left(R_{h}-u_{h},v_{h}\right)-d\left(v_{h},Q_{h}-p_{h}\right)-d\left(R_{h}-u_{h},q_{h}\right)+{\left(B\left(R_{h}-u_{h},u\right),v_{h}\right)}_{\Omega }+{\left(B\left(u_{h},R_{h}-u_{h}\right),v_{h}\right)}_{\Omega }\\ & = & {\left(B\left(R_{h}-u,u\right),v_{h}\right)}_{\Omega }+{\left(B\left(u_{h},R_{h}-u\right),v_{h}\right)}_{\Omega }\;\;\forall \left(v_{h},q_{h}\right)\in X_{h}\times M_{h}, \end{array}$$
where $\left(R_{h},Q_{h}\right)=\left(R_{h}\left(u,p\right),Q_{h}\left(u,p\right)\right)$. Using (Equation 118) with $q_{h}=0$, (Equation 31), (Equation 68) and the uniqueness condition 0<σ<1, we deduce
(119)$$\begin{array}{ccc} & & \beta_{0}∥Q_{h}-p_{h}{∥}_{0,\Omega }\leq \nu ∥R_{h}-u_{h}{∥}_{X}+N∥R_{h}-u_{h}{∥}_{X}{(∥u∥}_{X}+∥u_{h}{∥}_{X})\\ & + & N∥R_{h}{-u∥}_{0,\Omega }^{\frac{1}{2}}∥R_{h}{-u∥}_{X}^{\frac{1}{2}}{(∥u∥}_{X}+∥u_{h}{∥}_{X})\\ & \leq & 3\nu ∥R_{h}-u_{h}{∥}_{X}+2\sigma c_{6}h^{\frac{3}{2}}{(\nu ∥u∥}_{2,\Omega }+{∥p∥}_{1,\Omega })\\ & \leq & 3\nu ∥R_{h}-u_{h}{∥}_{X}+2\sigma c_{6}h^{\frac{3}{2}}C_{0}{∥F∥}_{0,\Omega }. \end{array}$$Next, taking $\left(v_{h},q_{h}\right)=\left(R_{h}-u_{h},-Q_{h}+p_{h}\right)$ in (Equation 118), using (Equation 119), (Equation 65) and (Equation 66), (Equation 30) and (Equation 31) and (Equation 37), we deduce
(120)$$\begin{array}{ccc} & & \nu \left(1-\sigma \right)∥R_{h}-u_{h}{∥}_{X}^{2}=-{\left(B\left(u-R_{h},u\right),R_{h}-u_{h}\right)}_{\Omega }-{\left(B\left(u_{h},u-R_{h}\right),R_{h}-u_{h}\right)}_{\Omega }\\ & \leq & 0.5∥u-R_{h}{∥}_{L^{3}}(∥\tilde{\nabla }{u∥}_{0,\Omega }∥R_{h}-u_{h}{∥}_{L^{6}}+∥\tilde{\nabla }\left(R_{h}u-u_{h}\right){∥}_{0,\Omega }{∥u∥}_{L^{6}})\\ & + & ∥u-R_{h}{∥}_{L^{3}}(∥u_{h}{∥}_{L^{6}}∥\tilde{\nabla }\left(R_{h}-u_{h}\right){∥}_{0,\Omega }+∥\tilde{\nabla }u_{h}{∥}_{0,\Omega }∥R_{h}u-u_{h}{∥}_{L^{6}})\\ & \leq & 2c_{1}^{\frac{3}{2}}∥u-R_{h}{∥}_{0,\Omega }^{\frac{1}{2}}∥u-R_{h}{∥}_{X}^{\frac{1}{2}}{(∥u∥}_{X}+∥u_{h}{∥}_{X})∥R_{h}-u_{h}{∥}_{X}\\ & \leq & 2c_{1}^{\frac{3}{2}}c_{6}h^{\frac{3}{2}}\nu^{-1}{(\nu ∥u∥}_{2,\Omega }+{∥p∥}_{1,\Omega }{)(∥u∥}_{x}+∥u_{h}{∥}_{X})∥R_{h}-u_{h}{∥}_{X}\\ & \leq & Nc_{6}h^{\frac{3}{2}}\nu^{-2}{(\nu ∥u∥}_{2,\Omega }+{∥p∥}_{1,\Omega }{)(\nu ∥u∥}_{x}+\nu ∥u_{h}{∥}_{X})∥R_{h}-u_{h}{∥}_{X}\\ & \leq & 2\sigma c_{6}h^{\frac{3}{2}}C_{0}{∥F∥}_{0,\Omega }{∥R_{h}-u_{h}∥}_{X}. \end{array}$$Combining (Equation 119) and (Equation 120) and (Equation 65) and (Equation 66), we deduce
(121)$$\begin{array}{c} \nu ∥R_{h}-u_{h}{∥}_{X}\leq 2c_{6}C_{0}h^{\frac{3}{2}}\frac{\sigma }{1-\sigma }{∥F∥}_{0,\Omega },\;\;\beta_{0}∥Q_{h}-p_{h}{∥}_{0,\Omega }\leq 5c_{6}C_{0}h^{\frac{3}{2}}\frac{\sigma }{1-\sigma }{∥F∥}_{0,\Omega }. \end{array}$$Combining (Equation 121) with (Equation 115) and using the stability condition $\sigma h^{\frac{1}{2}}\leq 1-\sigma$, we obtain
(122)$$\begin{array}{c} \nu ∥u-u_{h}{∥}_{X}\leq {Ch∥F∥}_{0,\Omega },\;\;\beta_{0}∥p-p_{h}{∥}_{0,\Omega }\leq Ch{∥F∥}_{0,\Omega }. \end{array}$$In order to estimate the $L^{2}\left(\Omega \right)$ bound of the error estimate $u-u_{h}$, we consider the dual equation of the 3D Navier–Stokes equations with the Dirichlet boundary condition
(123)$$\begin{array}{ccc} -\nu \tilde{\Delta }\phi +\tilde{\nabla }\psi +B^{\prime }\left(u,\phi \right)-B\left(u_{h},\phi \right) & = & u-u_{h},\;\mathrm{in}\Omega , \end{array}$$
(124)$$\begin{array}{ccc} \tilde{\nabla }\cdot \phi & = & 0,\;\;\;\;\mathrm{in}\Omega , \end{array}$$
(125)$$\begin{array}{ccc} \phi & = & 0,\;\;\;\;\mathrm{on}\partial \Omega , \end{array}$$
where ${\left(B\left(v,u\right),\phi \right)}_{\Omega }={\left(v,B^{\prime }\left(u,\phi \right)\right)}_{\Omega }$.Using the Green formula, we deduce the weak formulation of the dual Equations (Equation 123) and (Equation 125): Find (ϕ,ψ)∈X×M such that for each (v,q)∈X×M there holds
(126)$$\begin{array}{c} \nu A\left(\phi ,v\right)-d\left(v,\psi \right)-d\left(\phi ,q\right)+{\left(B\left(v,u\right),\phi \right)}_{\Omega }+{\left(B\left(u_{h},v\right),\phi \right)}_{\Omega }={\left(u-u_{h},v\right)}_{\Omega }. \end{array}$$Setting
$$\tilde{A}\left(\phi ,v\right)=\nu A\left(\phi ,v\right)+{\left(B\left(v,u\right),\phi \right)}_{\Omega }+{\left(B\left(u_{h},v\right),\phi \right)}_{\Omega },$$
we deduce from (Equation 30) and (Equation 31) that
(127)$$\begin{array}{ccc} & & |\tilde{A}{\left(\phi ,v\right)|\leq \nu ∥\phi ∥}_{X}{∥v∥}_{X}+{N∥v∥}_{X}{∥\phi ∥}_{X}{(∥u∥}_{X}+∥u_{h}{∥}_{X}{)\leq 3\nu ∥\phi ∥}_{X}{∥v∥}_{X}, \end{array}$$
(128)$$\begin{array}{ccc} & & \tilde{A}\left(\phi ,\phi \right)=\nu A\left(\phi ,\phi \right)+{\left(B\left(\phi ,u\right),\phi \right)}_{\Omega }\geq \left(1-\sigma \right)\nu {∥\phi ∥}_{X}^{2}. \end{array}$$Using Lemma 3 with $A^{n-1}\left(u,v\right)=\tilde{A}\left(\phi ,v\right)$, we show that (Equation 126) admits a unique solution (ϕ,ψ)∈X×M. Using (Equation 126)–(Equation 128) and Lemma 1, we have
(129)$$\begin{array}{ccc} & & {\nu \left(1-\sigma \right)∥\phi ∥}_{X}\leq {∥u-u_{h}∥}_{-1,\Omega }, \end{array}$$
(130)$$\begin{array}{ccc} & & {\beta ∥\psi ∥}_{0,\Omega }\leq ∥u-u_{h}{∥}_{-1,\Omega }+{3\nu ∥\phi ∥}_{X}\leq \left(1+\frac{3}{1-\sigma }\right){∥u-u_{h}∥}_{-1,\Omega }. \end{array}$$Using again (Equation 25), (Equation 30)–(Equation 32) and (Equation 126), we have
(131)$$\begin{array}{ccc} & & {\nu ∥\phi ∥}_{2,\Omega }+{∥\psi ∥}_{1,\Omega }\leq c_{0}∥u-u_{h}{∥}_{0,\Omega }+c_{0}∥B^{\prime }{\left(u,\phi \right)∥}_{0,\Omega }+c_{0}{∥B\left(u_{h},\phi \right)∥}_{0,\Omega }\\ & \leq & c_{0}∥u-u_{h}{∥}_{0,\Omega }+2c_{0}{N(∥u∥}_{X}+∥u_{h}{∥}_{X}{)∥\phi ∥}_{X}^{\frac{1}{2}}{∥\phi ∥}_{2,\Omega }^{\frac{1}{2}}\\ & \leq & \frac{1}{2}{\nu ∥\phi ∥}_{2,\Omega }+c_{0}∥u-u_{h}{∥}_{0,\Omega }+4c_{0}^{2}N^{2}\nu^{-1}{(∥u∥}_{X}^{2}+∥u_{h}{∥}_{X}^{2}{)∥\phi ∥}_{X}\\ & \leq & \frac{1}{2}{\nu ∥\phi ∥}_{2,\Omega }+c_{0}∥u-u_{h}{∥}_{0,\Omega }+4c_{0}^{2}N^{2}\nu^{-4}{∥F∥}_{-1,\Omega }^{2}\nu {∥\phi ∥}_{X}\\ & \leq & \frac{1}{2}{\nu ∥\phi ∥}_{2,\Omega }+c_{0}∥u-u_{h}{∥}_{0,\Omega }+4c_{0}^{2}\gamma_{\Omega }\frac{1}{1-\sigma }{∥u-u_{h}∥}_{0,\Omega }, \end{array}$$
which yields
(132)$$\begin{array}{ccc} & & {\nu ∥\phi ∥}_{2,\Omega }+{∥\psi ∥}_{1,\Omega }\leq 2c_{0}∥u-u_{h}{∥}_{0,\Omega }+8c_{0}^{2}\gamma_{\Omega }{\left(1-\sigma \right)}^{-1}{∥u-u_{h}∥}_{0,\Omega }. \end{array}$$Taking $\left(v,q\right)=\nu \left(u-u_{h},-p+p_{h}\right)$ in (Equation 126), using (Equation 118), (Equation 132) and (Equation 65) and (Equation 66), we deduce
(133)$$\begin{array}{ccc} & & \nu ∥u-u_{h}{∥}_{0,\Omega }^{2}=\nu^{2}A\left(u-u_{h},\phi \right)-\nu d\left(u-u_{h},\psi \right)-\nu d\left(\phi ,p-p_{h}\right)\\ & + & \nu {\left(B\left(u-u_{h},u\right),\phi \right)}_{\Omega }+\nu {\left(B\left(u_{h},u-u_{h}\right),\phi \right)}_{\Omega }\\ & = & \nu^{2}A\left(u-u_{h},\phi -\pi_{h}\phi \right)-\nu d\left(u-u_{h},\psi -\rho_{h}\psi \right)-\nu d\left(\phi -\pi_{h}\phi ,p-p_{h}\right)\\ & + & \nu {\left(B\left(u-u_{h},u\right),\phi -\pi_{h}\phi \right)}_{\Omega }+\nu {\left(B\left(u_{h},u-u_{h}\right),\phi -\pi_{h}\phi \right)}_{\Omega }\\ & \leq & \nu^{2}∥u-u_{h}{∥}_{X}∥\phi -\pi_{h}{\phi ∥}_{X}+\sqrt{3}(\nu ∥u-u_{h}{∥}_{X}∥\psi -\rho_{h}{\psi ∥}_{0,\Omega }+\nu ∥\phi -\pi_{h}{\phi ∥}_{X}∥p-p_{h}{∥}_{0,\Omega })\\ & + & N∥u-u_{h}{∥}_{X}{(∥u∥}_{X}+∥u_{h}{∥}_{X})\nu ∥\phi -\pi_{h}{\phi ∥}_{X}\\ & \leq & \sqrt{3}c_{3}h(\nu ∥u-u_{h}{∥}_{X}+∥p-p_{h}{∥}_{0,\Omega }{)(\nu ∥\phi ∥}_{2,\Omega }+{∥\psi ∥}_{1,\Omega })\\ & + & N\nu ∥u-u_{h}{∥}_{X}{(∥u∥}_{X}+∥u_{h}{∥}_{X})c_{3}h{∥\phi ∥}_{2,\Omega }\\ & \leq & \frac{C}{1-\sigma }h(\nu ∥u-u_{h}{∥}_{X}+∥p-p_{h}{∥}_{0,\Omega })∥u-u_{h}{∥}_{0,\Omega }. \end{array}$$Combining (Equation 133) with (Equation 122), we obtain (Equation 116).The proof ends. □

## 5. Oseen Iterative FE Method

Referring to the nonlinearity of the weak formulation (Equation 84), we design the Oseen iterative FE method of the 3D steady Navier–Stokes equations as follows. Setting $u_{h}^{-1}=0$, we define the Oseen iterative FE solution $\left(u_{h}^{n},p_{h}^{n}\right)$ of the 3D steady Navier–Stokes equations: find $\left(u_{h}^{n},p_{h}^{n}\right)\in X_{h}\times M_{h}$ such that for each $\left(v_{h},q_{h}\right)\in X_{h}\times M_{h}$ there holds
(134)$$\begin{array}{c} \nu A\left(u_{h}^{n},v_{h}\right)-d\left(v_{h},p_{h}^{n}\right)-d\left(u_{h}^{n},q_{h}\right)+{\left(B\left(u_{h}^{n-1},u_{h}^{n}\right),v_{h}\right)}_{\Omega }={\left(F,v_{h}\right)}_{\Omega }, \end{array}$$
or
(135)$$\begin{array}{c} A^{n-1}\left(u_{h}^{n},v_{h}\right)-d\left(v_{h},p_{h}^{n}\right)-d\left(u_{h}^{n},q_{h}\right)={\left(F,v_{h}\right)}_{\Omega }, \end{array}$$
where
$$\begin{array}{c} A^{n-1}\left(u_{h},v_{h}\right)=\nu A\left(u_{h},v_{h}\right)+{\left(B\left(u_{h}^{n-1},u_{h}\right),v_{h}\right)}_{\Omega }. \end{array}$$

In order to prove the existence, uniqueness and stability of the solution $\left(u_{h}^{n},p_{h}^{n}\right)$ based on (Equation 134) or (Equation 135), we consider the continuous and elliptic condition of the bilinear form $A^{n-1}\left(u_{h},v_{h}\right)$.

**Lemma** **10.**
*If $\nu ∥u_{h}^{n-1}{∥}_{X}\leq {∥F∥}_{-1,\Omega }$ and 0<σ<1, then the bilinear form $A^{n-1}\left(u_{h},v_{h}\right)$ satisfies*

(136)
$$\begin{array}{ccc} A^{n-1}\left(u_{h},v_{h}\right) & \leq & 2\nu ∥u_{h}{∥}_{X}{∥v_{h}∥}_{X}, \end{array}$$


(137)
$$\begin{array}{ccc} A^{n-1}\left(u_{h},u_{h}\right) & = & \nu ∥u_{h}{∥}_{X}^{2}, \end{array}$$

*for each uh,vh∈Xh.*


**Proof.** Using (Equation 30) and (Equation 31), we have
$$\begin{array}{ccc} A^{n-1}\left(u_{h},v_{h}\right) & \leq & \nu ∥u_{h}{∥}_{X}∥v_{h}{∥}_{X}+N∥u_{h}^{n-1}{∥}_{X}∥u_{h}{∥}_{X}∥v_{h}{∥}_{X}\leq 2\nu ∥u_{h}{∥}_{X}{∥v_{h}∥}_{X},\\ A^{n-1}\left(u_{h},u_{h}\right) & = & \nu ∥u_{h}{∥}_{X}^{2}+{\left(B\left(u_{h}^{n-1},u_{h}\right),u_{h}\right)}_{\Omega }=\nu {∥u_{h}∥}_{X}^{2}, \end{array}$$
which are (Equation 136) and (Equation 137). The proof ends. □

Furthermore, we obtain the existence, uniqueness, stability and convergence results of the solution $\left(u_{h}^{n},p_{h}^{n}\right)$ based on (Equation 134) or (Equation 135).

**Lemma** **11.**
*If $F\in H^{-1}{\left(\Omega \right)}^{3}$ and 0<σ<1, $X_{h}\times M_{h}$ satisfies (Equation 68), then (Equation 134) or (Equation 135) admits a unique solution $\left(u_{h}^{n},p_{h}^{n}\right)\in X_{h}\times M_{h}$ such that*

(138)
$$\begin{array}{c} \nu ∥u_{h}^{n}{∥}_{X}\leq {∥F∥}_{-1,\Omega },\;\;\beta_{0}∥p_{h}^{n}{∥}_{0,\Omega }\leq 3{∥F∥}_{-1,\Omega }, \end{array}$$

*and*

(139)
$$\begin{array}{ccc} \nu ∥u_{h}^{n}-u_{h}{∥}_{X}\leq \sigma^{n+1}{∥F∥}_{-1,\Omega },\;\;\beta_{0}{∥p_{h}^{n}-p_{h}∥}_{0,\Omega } & \leq & 3\sigma^{n+1}{∥F∥}_{-1,\Omega }. \end{array}$$



**Proof.** For n=0 and $u_{h}^{-1}=0$, we deduce from (Equation 134) that $\left(u_{h}^{0},p_{h}^{0}\right)\in X_{h}\times M_{h}$ satisfies
(140)$$\begin{array}{c} \nu A\left(u_{h}^{0},v_{h}\right)-d\left(v_{h},p_{h}^{0}\right)-d\left(\nu u_{h}^{0},q_{h}\right)={\left(F,v_{h}\right)}_{\Omega }, \end{array}$$
for each $\left(v_{h},q_{h}\right)\in X_{h}\times M_{h}$.Using Lemma 4, we show the existence and uniqueness of the solution $\left(u_{h}^{0},p_{h}^{0}\right)$ satisfying (Equation 138). Moreover, using (Equation 140) and (Equation 84), we easily show that $\left(u_{h}^{0},p_{h}^{0}\right)\in X_{h}\times M_{h}$ satisfies
(141)$$\begin{array}{c} \nu A\left(u_{h}^{0}-u_{h},v_{h}\right)-d\left(v_{h},p_{h}^{0}-p_{h}\right)-d\left(\nu u_{h}^{0}-\nu u_{h},q_{h}\right)=-{\left(B\left(u_{h},u_{h}\right),v_{h}\right)}_{\Omega }. \end{array}$$Using again Lemma 4, Lemma 6 and (Equation 141), we have
(142)$$\begin{array}{c} \nu ∥u_{h}^{0}-u_{h}{∥}_{X}\leq N∥u_{h}{∥}_{X}^{2}\leq {\sigma ∥F∥}_{-1,\Omega },\;\;\beta_{0}∥p_{h}^{0}-p_{h}∥\leq 2N∥u_{h}{∥}_{X}^{2}\leq 2\sigma {∥F∥}_{-1,\Omega }. \end{array}$$Thus, we show that Lemma 11 holds for n=0.Now, assuming that the conclusions of Lemma 11 hold for n−1, we want to prove that Lemma 11 holds for *n*. Using the induction assumption for n−1, Lemma 10 and Lemma 5, we deduce that (Equation 135) admits a unique solution $(u_{h}^{n},p_{h}^{n})\in X\times M$ which satisfies
(143)$$\begin{array}{ccc} & & \nu ∥u_{h}^{n}{∥}_{X}\leq {∥F∥}_{-1,\Omega },\;\;\beta_{0}∥p_{h}^{n}{∥}_{0,\Omega }\leq 3{∥F∥}_{-1,\Omega }. \end{array}$$Next, it follows from (Equation 134) and (Equation 84) that
(144)$$\begin{array}{ccc} & & \nu A\left(u_{h}^{n}-u_{h},v_{h}\right)+{\left(B\left(u_{h}^{n-1}-u_{h},u_{h}^{n}\right),v_{h}\right)}_{\Omega }+{\left(B\left(u_{h},u_{h}^{n}-u_{h}\right),v_{h}\right)}_{\Omega }\\ & & -d\left(v_{h},p_{h}^{n}-p_{h}\right)-d\left(u_{h}^{n}-u_{h},q_{h}\right)=0. \end{array}$$Taking $\left(v_{h},q_{h}\right)=\left(u_{h}^{n}-u_{h},-p_{h}^{n}+p_{h}\right)\in X_{h}\times M_{h}$ in (Equation 144) and using (Equation 30) and (Equation 31) yields
$$\begin{array}{ccc} & & \nu ∥u_{h}^{n}-u_{h}{∥}_{X}\leq N∥u_{h}^{n-1}-u_{h}{∥}_{X}{∥u_{h}^{n}∥}_{X}\\ & \leq & N\nu^{-2}{∥F∥}_{-1,\Omega }\nu ∥u_{h}^{n-1}-u_{h}{∥}_{X}\leq \sigma \nu {∥u_{h}^{n-1}-u_{h}∥}_{X}, \end{array}$$
which, with the induction assumption for n−1, yields
(145)$$\begin{array}{ccc} & & \nu ∥u_{h}^{n}-u_{h}{∥}_{X}\leq \sigma^{n+1}{∥F∥}_{-1,\Omega }. \end{array}$$Finally, using (Equation 31), (Equation 145) and Lemma 6, we obtain
(146)$$\begin{array}{ccc} & & \beta_{0}∥p_{h}^{n}-p_{h}{∥}_{0,\Omega }\leq \nu ∥u_{h}^{n}-u_{h}{∥}_{X}+N∥u_{h}^{n-1}-u_{h}{∥}_{X}∥u_{h}^{n}{∥}_{X}+N∥u_{h}^{n}-u_{h}{∥}_{X}{∥u_{h}∥}_{X}\\ & \leq & \nu ∥u_{h}^{n}-u_{h}{∥}_{X}+N\nu^{-2}{∥F∥}_{-1,\Omega }\nu (∥u_{h}^{n}-u_{h}{∥}_{X}+\nu ∥u_{h}^{n-1}-u_{h}{∥}_{X})\\ & \leq & 2\nu ∥u_{h}^{n}-u_{h}{∥}_{X}+\sigma \nu ∥u_{h}^{n-1}-u_{h}{∥}_{X}\leq 3\sigma^{n+1}{∥F∥}_{-1,\Omega }. \end{array}$$Combining (Equation 146) and (Equation 145) and using the induction assumption for n−1 yields (Equation 139). Hence, Lemma 11 holds for *n*. The proof ends. □

Finally, by combining Lemma 11 with Lemma 9, we obtain the convergence result of the Oseen iterative FE solution $\left(u_{h}^{n},p_{h}^{n}\right)$ with respect to the exact solution (u,p) of the 3D steady Navier–Stokes equations.

**Theorem** **3.**
*If $F\in L^{2}\left(\Omega \right)$ and $X_{h}\times M_{h}$ satisfies (Equation 65)–(Equation 68), 0<σ<1 and $\sigma h^{\frac{1}{2}}\leq 1-\sigma$, then the Oseen iterative FE solution $\left(u_{h}^{n},p_{h}^{n}\right)$ satisfies the following error estimates:*

(147)
$$\begin{array}{c} \nu ∥u-u_{h}^{n}{∥}_{X}\leq \sigma^{n+1}{∥F∥}_{-1,\Omega }+Ch{∥F∥}_{0,\Omega }, \end{array}$$


(148)
$$\begin{array}{c} \beta_{0}∥p-p_{h}^{n}{∥}_{0,\Omega }\leq 3\sigma^{n+1}{∥F∥}_{-1,\Omega }+Ch{∥F∥}_{0,\Omega }, \end{array}$$


(149)
$$\begin{array}{c} \;\;\nu ∥u-u_{h}^{n}{∥}_{0,\Omega }\leq \sigma^{n+1}\gamma_{\Omega }{∥F∥}_{-1,\Omega }+\frac{C}{1-\sigma }h^{2}{∥F∥}_{0,\Omega }. \end{array}$$



## 6. Newton Iterative FE Method

In this section, referring to the nonlinearity of the weak formulation (Equation 84), we design the Newton iterative FE method of the 3D steady Navier–Stokes equations as follows. Setting $u_{h}^{-1}=0$, we define the Newton iterative FE solution $\left(u_{h}^{n},p_{h}^{n}\right)$ of the 3D steady Navier–Stokes equations: find $\left(u_{h}^{n},p_{h}^{n}\right)\in X_{h}\times M_{h}$ such that for each $\left(v_{h},q_{h}\right)\in X_{h}\times M_{h}$ there holds
(150)$$\begin{array}{ccc} & & \nu A\left(u_{h}^{n},v_{h}\right)-d\left(v_{h},p_{h}^{n}\right)-d\left(u_{h}^{n},q_{h}\right)+{\left(B\left(u_{h}^{n-1},u_{h}^{n}\right),v_{h}\right)}_{\Omega }\\ & + & {\left(B\left(u_{h}^{n},u_{h}^{n-1}\right),v_{h}\right)}_{\Omega }-{\left(B\left(u_{h}^{n-1},u_{h}^{n-1}\right),v_{h}\right)}_{\Omega }={\left(F,v_{h}\right)}_{\Omega }, \end{array}$$
or
(151)$$\begin{array}{c} A^{n-1}\left(u_{h}^{n},v_{h}\right)-d\left(v_{h},p_{h}^{n}\right)-d\left(u_{h}^{n},q_{h}\right)=\tilde{F}\left(v_{h}\right), \end{array}$$
where
$$\begin{array}{ccc} & & \tilde{F}\left(v_{h}\right)={\left(F,v_{h}\right)}_{\Omega }+{\left(B\left(u_{h}^{n-1},u_{h}^{n-1}\right),v_{h}\right)}_{\Omega },\\ & & A^{n-1}\left(u_{h},v_{h}\right)=\nu A\left(u_{h},v_{h}\right)+{\left(B\left(u_{h}^{n-1},u_{h}\right),v_{h}\right)}_{\Omega }+{\left(B\left(u_{h},u_{h}^{n-1}\right),v_{h}\right)}_{\Omega }. \end{array}$$

In order to prove the existence, uniqueness and stability of the solution $\left(u_{h}^{n},p_{h}^{n}\right)$ based on (Equation 150) or (Equation 151), we consider the continuous of the linear form $\tilde{F}\left(v_{h}\right)$ and the continuous and elliptic condition of the bilinear form $A^{n-1}\left(u_{h},v_{h}\right)$.

**Lemma** **12.**
*If $\nu ∥u_{h}^{n-1}{∥}_{X}\leq \frac{6}{5}{∥F∥}_{-1,\Omega }$ and $0<\sigma \leq \frac{5}{11}$, then the bilinear form $A^{n-1}\left(u_{h},v_{h}\right)$ satisfies*

(152)
$$\begin{array}{ccc} & & |\tilde{F}\left(v_{h}\right){|\leq 2∥F∥}_{-1,\Omega }{∥v_{h}∥}_{X}, \end{array}$$


(153)
$$\begin{array}{ccc} & & A^{n-1}\left(u_{h},v_{h}\right)\leq 3\nu ∥u_{h}{∥}_{X}{∥v_{h}∥}_{X}, \end{array}$$


(154)
$$\begin{array}{ccc} & & A^{n-1}\left(u_{h},u_{h}\right)\geq \sigma \nu {∥u_{h}∥}_{X}^{2}, \end{array}$$

*for each uh,vh∈Xh.*


**Proof.** Using (Equation 30) and (Equation 31), we have
$$\begin{array}{ccc} & & |\tilde{F}\left(v_{h}\right){|\leq ∥F∥}_{-1,\Omega }∥v_{h}{∥}_{X}+N∥u_{h}^{n-1}{∥}_{X}^{2}∥v_{h}{∥}_{X}\leq {∥F∥}_{-1,\Omega }∥v_{h}{∥}_{X}\left(1+\sigma \frac{36}{25}\right)\leq {2∥F∥}_{-1,\Omega }{∥v_{h}∥}_{X},\\ & & A^{n-1}\left(u_{h},v_{h}\right)\leq \nu ∥u_{h}{∥}_{X}∥v_{h}{∥}_{X}+2N∥u_{h}^{n-1}{∥}_{X}∥u_{h}{∥}_{X}∥v_{h}{∥}_{X}\leq \frac{23}{11}\nu ∥u_{h}{∥}_{X}{∥v_{h}∥}_{X},\\ & & A^{n-1}\left(u_{h},u_{h}\right)=\nu ∥u_{h}{∥}_{X}^{2}+{\left(B\left(u_{h},u_{h}^{n-1}\right),u_{h}\right)}_{\Omega }\geq \sigma \nu {∥u_{h}∥}_{X}^{2}, \end{array}$$
which are (Equation 152)–(Equation 154). The proof ends. □

Furthermore, we obtain the existence, uniqueness, stability and convergence results of the solution $\left(u_{h}^{n},p_{h}^{n}\right)$ based on (Equation 150) or (Equation 151).

**Lemma** **13.**
*If $F\in H^{-1}{\left(\Omega \right)}^{3}$ and $0<\sigma \leq \frac{5}{11}$, $X_{h}\times M_{h}$ satisfies (Equation 68), then (Equation 150) or (Equation 151) admits a unique solution $\left(u_{h}^{n},p_{h}^{n}\right)\in X_{h}\times M_{h}$ such that*

(155)
$$\begin{array}{ccc} & & \nu ∥u_{h}^{n}{∥}_{X}\leq \frac{6}{5}{∥F∥}_{-1,\Omega }, \end{array}$$


(156)
$$\begin{array}{ccc} & & \beta_{0}∥p_{h}^{n}{∥}_{0,\Omega }\leq 3{∥F∥}_{-1,\Omega }, \end{array}$$

*and*

(157)
$$\begin{array}{ccc} & & \nu ∥u_{h}^{n}-u_{h}{∥}_{X}\leq \sigma^{2^{n}}{∥F∥}_{-1,\Omega }, \end{array}$$


(158)
$$\begin{array}{ccc} & & \beta_{0}∥p_{h}^{n}-p_{h}{∥}_{0,\Omega }\leq 3\sigma^{2^{n}}{∥F∥}_{-1,\Omega }. \end{array}$$



**Proof.** For n=0 and $u_{h}^{-1}=0$, we deduce from (Equation 150) that $\left(u_{h}^{0},p_{h}^{0}\right)\in X_{h}\times M_{h}$ satisfies
(159)$$\begin{array}{c} \nu A\left(u_{h}^{0},v_{h}\right)-d\left(v_{h},p_{h}^{0}\right)-d\left(\nu u_{h}^{0},q_{h}\right)={\left(F,v_{h}\right)}_{\Omega }, \end{array}$$
for each $\left(v_{h},q_{h}\right)\in X_{h}\times M_{h}$.Using Lemma 4, we show the existence and uniqueness of the solution $\left(u_{h}^{0},p_{h}^{0}\right)$ satisfying (Equation 155) and (Equation 156). Moreover, using (Equation 159) and (Equation 84), we easily show that $\left(u_{h}^{0},p_{h}^{0}\right)\in X_{h}\times M_{h}$ satisfies
(160)$$\begin{array}{c} \nu A\left(u_{h}^{0}-u_{h},v_{h}\right)-d\left(v_{h},p_{h}^{0}-p_{h}\right)-d\left(\nu u_{h}^{0}-\nu u_{h},q_{h}\right)=-{\left(B\left(u_{h},u_{h}\right),v_{h}\right)}_{\Omega }. \end{array}$$Using again Lemma 4, Lemma 6 and (Equation 160), we have
(161)$$\begin{array}{ccc} & & \nu ∥u_{h}^{0}-u_{h}{∥}_{X}\leq N∥u_{h}{∥}_{X}^{2}\leq \sigma {∥F∥}_{-1,\Omega }, \end{array}$$
(162)$$\begin{array}{ccc} & & \;\;\beta_{0}∥p_{h}^{0}-p_{h}∥\leq \nu ∥u_{h}^{0}-u_{h}{∥}_{X}+N∥u_{h}{∥}_{X}^{2}\leq 2\sigma {∥F∥}_{-1,\Omega }, \end{array}$$
which yield (Equation 157) and (Equation 158) for n=0. Thus, we show that Lemma 13 holds for n=0.Now, assuming that the conclusions of Lemma 13 hold for n−1, we want to prove that Lemma 13 holds for *n*.Next, we deduce from (Equation 150), (Equation 161), (Equation 30) and (Equation 31) and the induction assumption for n−1 that $\left(u_{h}^{n},p_{h}^{n}\right)$ and $\left(e^{n},\eta^{n}\right)=\left(u_{h}^{n}-u_{h},p_{h}^{n}-p_{h}\right)$ satisfy
(163)$$\begin{array}{ccc} & & \nu A\left(u_{h}^{n},v_{h}\right)-d\left(v_{h},p_{h}^{n}\right)-d\left(u_{h}^{n},q_{h}\right)+{\left(B\left(u_{h}^{n},u_{h}^{n}\right),v_{h}\right)}_{\Omega }-{\left(B\left(u_{h}^{n}-u_{h}^{n-1},u_{h}^{n}-u_{h}^{n-1}\right),v_{h}\right)}_{\Omega }=0, \end{array}$$
(164)$$\begin{array}{ccc} & & \nu A\left(u_{h}^{n},u_{h}^{n}\right)-{\left(B\left(u_{h}^{n}-u_{h}^{n-1},u_{h}^{n}-u_{h}^{n-1}\right),u_{h}^{n}\right)}_{\Omega }={\left(F,u_{h}^{n}\right)}_{\Omega }, \end{array}$$
(165)$$\begin{array}{c} \nu A\left(e^{n},v_{h}\right)-d\left(v_{h},\eta^{n}\right)-d\left(e^{n},q_{h}\right)+{\left(B\left(e^{n},u_{h}^{n-1}\right),v_{h}\right)}_{\Omega }\\ +{\left(B\left(u_{h}^{n-1},e^{n}\right),v_{h}\right)}_{\Omega }-{\left(B\left(e^{n-1},e^{n-1}\right),v_{h}\right)}_{\Omega }=0, \end{array}$$
which yield
(166)$$\begin{array}{c} \left(1-\sigma \right)\nu ∥e^{1}{∥}_{X}\leq \nu (1-N\nu^{-1}∥u_{h}^{0}{∥}_{X})∥e^{1}{∥}_{X}\\ \leq N∥e^{0}{∥}_{X}^{2}\leq N\nu^{-2}\sigma^{2}{∥F∥}_{-1,\Omega }^{2}\leq \sigma^{3}{∥F∥}_{-1,\Omega }\leq \left(1-\sigma \right)\frac{5}{6}\sigma^{2}{∥F∥}_{-1,\Omega }, \end{array}$$
(167)$$\begin{array}{c} \nu ∥u_{h}^{1}-u_{h}^{0}{∥}_{X}\leq \nu (∥e^{1}{∥}_{X}+∥e^{0}{∥}_{X})\leq \left(1+\frac{5}{6}\sigma \right){\sigma ∥F∥}_{-1,\Omega }{∥F∥}_{-1,\Omega }, \end{array}$$
(168)$$\begin{array}{c} \nu ∥u_{h}^{1}{∥}_{X}\leq N∥u_{h}^{1}-u_{h}^{0}{∥}_{X}^{2}+{∥F∥}_{-1,\Omega }\\ \leq {∥F∥}_{-1,\Omega }+N\nu^{-2}{\left(1+\frac{5}{6}\sigma \right)}^{2}\sigma^{2}{∥F∥}_{-1,\Omega }^{2}\leq \left[1+{\left(1+\frac{5}{6}\sigma \right)}^{2}\sigma^{3}\right]{∥F∥}_{-1,\Omega }\leq \frac{6}{5}{∥F∥}_{-1,\Omega }, \end{array}$$
(169)$$\begin{array}{ccc} & & \beta_{0}∥p_{h}^{1}{∥}_{0,\Omega }\leq \nu ∥u_{h}^{1}{∥}_{X}+{∥F∥}_{-1,\Omega }+N∥u_{h}^{0}{∥}_{X}(∥u_{h}^{1}{∥}_{X}+∥u_{h}^{1}-u_{h}^{0}{∥}_{X}{)\leq 3∥F∥}_{-1,\Omega }, \end{array}$$
(170)$$\begin{array}{ccc} & & \beta_{0}∥\eta^{1}{∥}_{0,\Omega }\leq \nu ∥e^{1}{∥}_{X}+N(2∥u_{h}^{0}{∥}_{X}∥e^{1}{∥}_{X}+∥e^{0}{∥}_{X}^{2})\leq 3\sigma^{2}{∥F∥}_{-1,\Omega }, \end{array}$$
for n=1 and
$$\begin{array}{ccc} & & \sigma \nu ∥e^{n}{∥}_{X}\leq \nu (1-N\nu^{-1}∥u_{h}^{n-1}{∥}_{X})∥e^{n}{∥}_{X} \end{array}$$
(171)$$\begin{array}{ccc} & \leq & N∥e^{n-1}{|}_{X}^{2}\leq N\nu^{-2}{\left(\sigma^{2^{n-1}}\right)}^{2}{∥F∥}_{-1,\Omega }^{2}\leq \sigma \sigma^{2^{n}}{∥F∥}_{-1,\Omega }, \end{array}$$
(172)$$\begin{array}{ccc} & & \nu ∥u_{h}^{n}-u_{h}^{n-1}{∥}_{X}\leq \nu ∥e^{n}-e^{n-1}{∥}_{X}\leq \left(\sigma^{2^{n}}+\sigma^{2^{n-1}}\right){∥F∥}_{-1,\Omega },\\ & & \nu ∥u_{h}^{n}{∥}_{X}\leq N∥u_{h}^{n}-u_{h}^{n-1}{∥}_{X}^{2}+{∥F∥}_{-1,\Omega } \end{array}$$
(173)$$\begin{array}{ccc} & \leq & {∥F∥}_{-1,\Omega }+\sigma {\left(\sigma^{2^{n}}+\sigma^{2^{n-1}}\right)}^{2}{∥F∥}_{-1,\Omega }\leq \frac{6}{5}{∥F∥}_{-1,\Omega }, \end{array}$$
(174)$$\begin{array}{ccc} & & \beta_{0}∥p_{h}^{n}{∥}_{0,\Omega }\leq \nu ∥u_{h}^{n}{∥}_{X}+{∥F∥}_{-1,\Omega }+N∥u_{h}^{n-1}{∥}_{X}(∥u_{h}^{n}{∥}_{X}+∥u_{h}^{n}-u_{h}^{n-1}{∥}_{X}{)\leq 3∥F∥}_{-1,\Omega }, \end{array}$$
(175)$$\begin{array}{ccc} & & \beta_{0}∥\eta^{n}{∥}_{0,\Omega }\leq \nu ∥e^{n}{∥}_{X}+2N∥u_{h}^{n-1}{∥}_{X}∥e^{n}{∥}_{X}+N∥e^{n-1}{∥}_{X}^{2}\leq 3\sigma^{2^{n}}{∥F∥}_{-1,\Omega }, \end{array}$$
for n≥2. Thus, (Equation 166)–(Equation 175) shows that (Equation 155)–(Equation 158) hold for *n*.The proof ends. □

Finally, by combining Lemma 13 with Lemma 9, we obtain the convergence result of the Newton iterative FE solution $\left(u_{h}^{n},p_{h}^{n}\right)$ with respect to the exact solution (u,p) of the 3D steady Navier–Stokes equations.

**Theorem** **4.**
*If $F\in L^{2}\left(\Omega \right)$ and $X_{h}\times M_{h}$ satisfies (Equation 65)–(Equation 68) and $0<\sigma \leq \frac{5}{11}$, then the Newton iterative FE solution $\left(u_{h}^{n},p_{h}^{n}\right)$ satisfies the following error estimates:*

(176)
$$\begin{array}{c} \nu ∥u-u_{h}^{n}{∥}_{X}\leq \sigma^{2^{n}}{∥F∥}_{-1,\Omega }+Ch{∥F∥}_{0,\Omega }, \end{array}$$


(177)
$$\begin{array}{c} \beta_{0}∥p-p_{h}^{n}{∥}_{0,\Omega }\leq 3\sigma^{2^{n}}{∥F∥}_{-1,\Omega }+Ch{∥F∥}_{0,\Omega }, \end{array}$$


(178)
$$\begin{array}{c} \;\;\nu ∥u-u_{h}^{n}{∥}_{0,\Omega }\leq \sigma^{2^{n}}\gamma_{\Omega }{∥F∥}_{-1,\Omega }+Ch^{2}{∥F∥}_{0,\Omega }. \end{array}$$



## 7. Stokes Iterative FE Method

In this section, referring to the nonlinearity of the weak formulation (Equation 84), we design the Stokes iterative FE method of the 3D steady Navier–Stokes equations as follows. Setting $u_{h}^{-1}=0$, we define the Stokes iterative FE solution $\left(u_{h}^{n},p_{h}^{n}\right)$ of the 3D steady Navier–Stokes equations: find $\left(u_{h}^{n},p_{h}^{n}\right)\in X_{h}\times M_{h}$ such that for each $\left(v_{h},q_{h}\right)\in X_{h}\times M_{h}$ there holds
(179)$$\begin{array}{c} \nu A\left(u_{h}^{n},v_{h}\right)-d\left(v_{h},p_{h}^{n}\right)-d\left(u_{h}^{n},q_{h}\right)+{\left(B\left(u_{h}^{n-1},u_{h}^{n-1}\right),v_{h}\right)}_{\Omega }={\left(F,v_{h}\right)}_{\Omega }, \end{array}$$
or
(180)$$\begin{array}{c} A^{n-1}\left(u_{h}^{n},v_{h}\right)-d\left(v_{h},p_{h}^{n}\right)-d\left(u_{h}^{n},q_{h}\right)=\tilde{F}\left(v_{h}\right), \end{array}$$
where
$$\begin{array}{c} \tilde{F}\left(v_{h}\right)={\left(F,v_{h}\right)}_{\Omega }-{\left(B\left(u_{h}^{n-1},u_{h}^{n-1}\right),v_{h}\right)}_{\Omega },\;\;A^{n-1}\left(u_{h},v_{h}\right)=\nu A\left(u_{h},v_{h}\right). \end{array}$$

In order to prove the existence, uniqueness and stability of the solution $\left(u_{h}^{n},p_{h}^{n}\right)$ based on (Equation 179) or (Equation 180), we consider the continuous condition of the linear form $\tilde{F}\left(v_{h}\right)$ and the continuous and elliptic conditions of the bilinear form $A^{n-1}\left(u_{h},v_{h}\right)$.

**Lemma** **14.**
*If $\nu ∥u_{h}^{n-1}{∥}_{X}\leq \frac{6}{5}{∥F∥}_{-1,\Omega }$ and $0<\sigma \leq \frac{2}{5}$, then the bilinear form $A^{n-1}\left(u_{h},v_{h}\right)$ satisfies*

(181)
$$\begin{array}{ccc} & & |\tilde{F}\left(v_{h}\right){|\leq 2∥F∥}_{-1,\Omega }{∥v_{h}∥}_{X}, \end{array}$$


(182)
$$\begin{array}{ccc} & & A^{n-1}\left(u_{h},v_{h}\right)\leq \nu ∥u_{h}{∥}_{X}{∥v_{h}∥}_{X}, \end{array}$$


(183)
$$\begin{array}{ccc} & & A^{n-1}\left(u_{h},u_{h}\right)=\nu {∥u_{h}∥}_{X}^{2}, \end{array}$$

*for each uh,vh∈Xh.*


**Proof.** The proof of (Equation 181)–(Equation 183) is very simple and can be omitted.Furthermore, we obtain the existence, uniqueness, stability and convergence results of the solution $\left(u_{h}^{n},p_{h}^{n}\right)$ based on (Equation 179) or (Equation 180). □

**Lemma** **15.**
*If $F\in H^{-1}{\left(\Omega \right)}^{3}$ and $0<\sigma \leq \frac{2}{5}$, $X_{h}\times M_{h}$ satisfy (Equation 68), then (Equation 179) or (Equation 180) admits a unique solution $\left(u_{h}^{n},p_{h}^{n}\right)\in X_{h}\times M_{h}$ such that*

(184)
$$\begin{array}{ccc} & & \nu ∥u_{h}^{n}{∥}_{X}\leq \frac{6}{5}{∥F∥}_{-1,\Omega },\;\;\beta_{0}∥p_{h}^{n}{∥}_{0,\Omega }\leq 3{∥F∥}_{-1,\Omega }, \end{array}$$


(185)
$$\begin{array}{ccc} & & \nu ∥u_{h}^{n}-u_{h}^{n-1}{∥}_{X}\leq {\left(\frac{12}{5}\sigma \right)}^{n-1}\sigma {∥F∥}_{-1,\Omega }\;\;\;\left(n\geq 1\right), \end{array}$$

*and*

(186)
$$\begin{array}{ccc} & & \nu ∥u_{h}^{n}-u_{h}{∥}_{X}\leq {\left(\frac{11}{5}\sigma \right)}^{n}\sigma {∥F∥}_{-1,\Omega }, \end{array}$$


(187)
$$\begin{array}{ccc} & & \beta_{0}∥p_{h}^{n}-p_{h}{∥}_{0,\Omega }\leq 3{\left(\frac{11}{5}\sigma \right)}^{n}\sigma {∥F∥}_{-1,\Omega }. \end{array}$$



**Proof.** For n=0 and $u_{h}^{-1}=0$, we deduce from (Equation 179) that $\left(u_{h}^{0},p_{h}^{0}\right)\in X_{h}\times M_{h}$ satisfies
(188)$$\begin{array}{c} \nu A\left(u_{h}^{0},v_{h}\right)-d\left(v_{h},p_{h}^{0}\right)-d\left(\nu u_{h}^{0},q_{h}\right)={\left(F,v_{h}\right)}_{\Omega }, \end{array}$$
for each $\left(v_{h},q_{h}\right)\in X_{h}\times M_{h}$.Using Lemma 4, we show the existence and uniqueness of the solution $\left(u_{h}^{0},p_{h}^{0}\right)$ satisfying (Equation 184) and (Equation 185). Moreover, using (Equation 188) and (Equation 84), we easily show that $\left(u_{h}^{0},p_{h}^{0}\right)\in X_{h}\times M_{h}$ satisfies
(189)$$\begin{array}{c} \nu A\left(u_{h}^{0}-u_{h},v_{h}\right)-d\left(v_{h},p_{h}^{0}-p_{h}\right)-d\left(\nu u_{h}^{0}-\nu u_{h},q_{h}\right)=-{\left(B\left(u_{h},u_{h}\right),v_{h}\right)}_{\Omega }. \end{array}$$Using again Lemma 4, Lemma 6 and (Equation 189), we have
(190)$$\begin{array}{ccc} & & \nu ∥u_{h}^{0}-u_{h}{∥}_{X}\leq N∥u_{h}{∥}_{X}^{2}\leq \sigma {∥F∥}_{-1,\Omega }, \end{array}$$
(191)$$\begin{array}{ccc} & & \;\;\beta_{0}∥p_{h}^{0}-p_{h}∥\leq \nu ∥u_{h}^{0}-u_{h}{∥}_{X}+N∥u_{h}{∥}_{X}^{2}\leq 2\sigma {∥F∥}_{-1,\Omega }. \end{array}$$Thus, we show that Lemma 15 holds for n=0.Now, assuming that the conclusions of Lemma 15 hold for 0,⋯,n−1, we want to prove that Lemma 15 holds for *n*.Next, we deduce from (Equation 179), (Equation 190), (Equation 30) and (Equation 31) and the induction assumption for 0,⋯,n−1 that $\left(u_{h}^{n},p_{h}^{n}\right)$ and $\left(e^{n},\eta^{n}\right)=\left(u_{h}^{n}-u_{h},p_{h}^{n}-p_{h}\right)$ satisfy
$$\begin{array}{ccc} & & \nu A\left(u_{h}^{n}-u_{h}^{n-1},v_{h}\right)-d\left(v_{h},p_{h}^{n}-p_{h}^{n-1}\right)-d\left(u_{h}^{n}-u_{h}^{n-1},q_{h}\right) \end{array}$$
(192)$$\begin{array}{ccc} & + & {\left(B\left(u_{h}^{n-1}-u_{h}^{n-2},u_{h}^{n-1}\right),v_{h}\right)}_{\Omega }+{\left(B\left(u_{h}^{n-2},u_{h}^{n-1}-u_{h}^{n-2}\right),v_{h}\right)}_{\Omega }=0, \end{array}$$
(193)$$\begin{array}{ccc} & & \nu A\left(u_{h}^{n},u_{h}^{n}\right)-{\left(B\left(u_{h}^{n-1},u_{h}^{n}-u_{h}^{n-1}\right),u_{h}^{n}\right)}_{\Omega }={\left(F,u_{h}^{n}\right)}_{\Omega }, \end{array}$$
(194)$$\begin{array}{ccc} & & \nu A\left(e^{n},v_{h}\right)-d\left(v_{h},\eta^{n}\right)-d\left(e^{n},q_{h}\right)+{\left(B\left(e^{n-1},u_{h}^{n-1}\right),v_{h}\right)}_{\Omega }+{\left(B\left(u_{h},e^{n-1}\right),v_{h}\right)}_{\Omega }=0, \end{array}$$
which yield
$$\begin{array}{ccc} & & \nu ∥e^{1}{∥}_{X}\leq N(∥u_{h}^{0}{∥}_{X}+∥u_{h}{∥}_{X})∥e^{0}{∥}_{X} \end{array}$$
(195)$$\begin{array}{ccc} & \leq & 2N\nu^{-2}{∥F∥}_{-1,\Omega }∥e^{0}{∥}_{X}\leq 2\sigma^{2}{∥F∥}_{-1,\Omega }, \end{array}$$
(196)$$\begin{array}{ccc} & & \nu ∥u_{h}^{1}-u_{h}^{0}{∥}_{X}\leq N∥u_{h}^{0}{∥}_{X}^{2}\leq \sigma {∥F∥}_{-1,\Omega },\\ & & \nu ∥u_{h}^{1}{∥}_{X}\leq N∥u_{h}^{1}-u_{h}^{0}{∥}_{X}∥u_{h}^{0}{∥}_{X}+{∥F∥}_{-1,\Omega }\leq {∥F∥}_{-1,\Omega }+N\nu^{-2}\sigma {∥F∥}_{-1,\Omega }^{2} \end{array}$$
(197)$$\begin{array}{ccc} & \leq & \left(1+\sigma^{2}\right){∥F∥}_{-1,\Omega }\leq \frac{6}{5}{∥F∥}_{-1,\Omega }, \end{array}$$
(198)$$\begin{array}{ccc} & & \beta_{0}∥p_{h}^{1}{∥}_{0,\Omega }\leq \nu ∥u_{h}^{1}{∥}_{X}+{∥F∥}_{-1,\Omega }+N∥u_{h}^{0}{∥}_{X}^{2}\leq 3{∥F∥}_{-1,\Omega }, \end{array}$$
(199)$$\begin{array}{ccc} & & \beta_{0}∥\eta^{1}{∥}_{0,\Omega }\leq \nu ∥e^{1}{∥}_{X}+N(∥u_{h}^{0}{∥}_{X}+∥u_{h}{∥}_{X})∥e^{0}{∥}_{X}\leq 3\sigma^{2}{∥F∥}_{-1,\Omega }, \end{array}$$
for n=1 and
$$\begin{array}{ccc} & & \nu ∥u_{h}^{n}-u_{h}^{n-1}{∥}_{X}\leq N(∥u_{h}^{n-1}{∥}_{X}+∥u_{h}^{n-2}{∥}_{X})∥u_{h}^{n-1}-u_{h}^{n-2}{∥}_{X} \end{array}$$
(200)$$\begin{array}{ccc} & \leq & \frac{12}{5}\sigma ∥u_{h}^{n-1}-u_{h}^{n-2}{∥}_{X}\leq {\left(\frac{12}{5}\sigma \right)}^{n-1}\sigma {∥F∥}_{-1,\Omega },\\ & & \nu ∥u^{n}{∥}_{X}\leq N∥u_{h}^{n-1}{∥}_{X}∥u_{h}^{n}-u_{h}^{n-1}{∥}_{X}+{∥F∥}_{-1,\Omega } \end{array}$$
(201)$$\begin{array}{ccc} & \leq & {∥F∥}_{-1,\Omega }+\frac{6}{5}\sigma {\left(\frac{12}{5}\sigma \right)}^{n-1}{\sigma ∥F∥}_{-1,\Omega }\leq \frac{6}{5}{∥F∥}_{-1,\Omega }, \end{array}$$
(202)$$\begin{array}{ccc} & & \beta_{0}∥p_{h}^{n}{∥}_{0,\Omega }\leq \nu ∥u_{h}^{n}{∥}_{X}+{∥F∥}_{-1,\Omega }+N∥u_{h}^{n-1}{∥}_{X}^{2}\leq 3{∥F∥}_{-1,\Omega }, \end{array}$$
(203)$$\begin{array}{ccc} & & \nu ∥e^{n}{∥}_{X}\leq N(∥u_{h}{∥}_{X}+∥u_{h}^{n-1}{∥}_{X})∥e^{n-1}{∥}_{X}\leq \frac{11}{5}\sigma ∥e^{n-1}{∥}_{X}\leq {\left(\frac{11}{5}\sigma \right)}^{n}\sigma {∥F∥}_{-1,\Omega }, \end{array}$$
(204)$$\begin{array}{ccc} & & \beta_{0}∥\eta^{n}{∥}_{0,\Omega }\leq \nu ∥e^{n}{∥}_{X}+2N(∥u_{h}^{n-1}{∥}_{X}+∥u_{h}{∥}_{X})∥e^{n-1}{∥}_{X}\leq 3{\left(\frac{11}{5}\sigma \right)}^{n}\sigma {∥F∥}_{-1,\Omega }, \end{array}$$
for n≥2. Thus, (Equation 195)–(Equation 204) shows that (Equation 184)–(Equation 187) hold for *n*.The proof ends. □

Finally, by combining Lemma 15 with Lemma 9, we obtain the convergence result of the Stokes iterative FE solution $\left(u_{h}^{n},p_{h}^{n}\right)$ with respect to the exact solution (u,p) of the 3D steady Navier–Stokes equations.

**Theorem** **5.**
*If $F\in L^{2}\left(\Omega \right)$ and $X_{h}\times M_{h}$ satisfies (Equation 65)–(Equation 68) and $0<\sigma \leq \frac{2}{5}$, then the Stokes FE solution $\left(u_{h}^{n},p_{h}^{n}\right)$ satisfies the following error estimates:*

(205)
$$\begin{array}{c} \nu ∥u-u_{h}^{n}{∥}_{X}\leq {\left(\frac{11}{5}\sigma \right)}^{n}{\sigma ∥F∥}_{-1,\Omega }+Ch{∥F∥}_{0,\Omega }, \end{array}$$


(206)
$$\begin{array}{c} \beta_{0}∥p-p_{h}^{n}{∥}_{0,\Omega }\leq 3{\left(\frac{11}{5}\sigma \right)}^{n}{\sigma ∥F∥}_{-1,\Omega }+Ch{∥F∥}_{0,\Omega }, \end{array}$$


(207)
$$\begin{array}{c} \;\;\nu ∥u-u_{h}^{n}{∥}_{0,\Omega }\leq {\left(\frac{11}{5}\sigma \right)}^{n}\sigma \gamma_{\Omega }{∥F∥}_{-1,\Omega }+Ch^{2}{∥F∥}_{0,\Omega }. \end{array}$$



## 8. Conclusions

**Remark** **2.**
*From Theorems 3–5, we find that the three iterative FE methods are $H^{1}$-uniform stable and first-order convergent with respect to (ν,1−σ); the three iterative FE methods are $L^{2}$-uniform stable and second-order convergent with respect to ν; the Stokes iterative FE method is simpler than the Oseen iterative FE method; the Oseen iterative FE method is simpler than the Newton iterative FE method; the convergence of the Newton iterative FE method is better than that of the Oseen iterative FE method; and the convergence of the Oseen finite element iterative method is better than the Stokes iterative FE method in the case of $0<\sigma \leq \frac{2}{5}$.*


**Remark** **3.**
*From Theorems 3 and 4, we find that the two iterative FE methods are $H^{1}$-uniform stable and convergent with respect to (ν,1−σ); the three iterative FE methods are $L^{2}$-uniform stable and second-order convergent with respect to ν; the Oseen iterative FE method is simpler than the Newton iterative FE method; and the convergence of the Newton iterative FE method is better than the the Oseen iterative FE method in the case of $\frac{2}{5}<\sigma \leq \frac{5}{11}$.*


**Remark** **4.**
*From Theorem 3, we find that the Oseen iterative FE method is $H^{1}$-uniform stable and convergent with respect to (ν,1−σ) and the Oseen iterative FE method is $L^{2}$-uniform stable and second-order convergent with respect to ν in the case of $\frac{5}{11}<\sigma <1$.*


## Data Availability

Not applicable.

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
