# Peer review of "Finite Element Iterative Methods for the 3D Steady Navier--Stokes Equationsâ€"

_entropy, 2021, doi:10.3390/e23121659_

Round 1
Reviewer 1 Report
This is a short review of the manuscript "Finite element iterative methods for the 3D steady Navier-stokes equations". The manuscript presents weak formulations for 3D steady Stokes, Newton and Oseen equations as well as proof of existence, uniqueness and convergence of solutions.
I am I fear not competent to give a proper opinion on the mathematical aspect of the manuscript. It seems adequate however. I have few comments on the form:
- line 60 "p the pressure with and ..." does not make sense
- line 71 "General Lax-Miligram" => Milgram is more appropriate
- Line 111 : "Nowwe" => Now we
- Line 118 : Proof may be written in bold
- Line 176 : span^b(K) may be correctly spaced
- Equation 3.28 is very long and collides the equation number
Author Response
We have revised our manuscript based on the reviewer1's comments.

Reviewer 2 Report
This paper mainly investigated A finite element method based on finite element space pair XhXMh for 3D steady NS eqs. The general approach may be feasible, the proofs and the derivations may be correct. And the discussion is reasonable. It is suggested to accept this paper. A few things can be rectified before publication.
- improve the introduction part, add more information related to steady NS eqs under the uniqueness condition.
- In the introduction part, summarize the workflow of this study.
- Many variables with different symbols and notations are used in this paper. it is suggested to add a nomenclature part
- any numerical results?
Author Response
We have revised our manuscript based on the reviewer2's comments.

Reviewer 3 Report
Technical Comments
- “Lots of works are devoted to this problem, and the finite element methods, finite volume methods and finite difference methods have been the most successful methods. There are many scholars who have studied the numerical methods of the Navier-Stokes equations; see, e.g., the monograph………………..”
An important area that is left out is the development of high order spectral volume and spectral difference methods advanced by Kannan et al [1-8]. These need to be added to the manuscript.
- Remark 8.1: 0 < σ ≤ 2 5. Can you generalize this for 1D and 2D cases?
- Similarly remark 8:2: 5 < σ ≤ 11 5 .
Can you generalize this to 1D and 2D cases?
Author Response

(The authors gave the same response as above.)
